# Dissociable roles of distinct thalamic circuits in learning reaches to spatial targets

Leslie J. Sibener [1,4] ✉, Alice C. Mosberger [1,4], Tiffany X. Chen[1], Vivek R. Athalye [1], James M. Murray [2] & Rui M. Costa[1,3] ✉

Reaching movements are critical for survival, and are learned and controlled by distributed motor networks. Even though the thalamus is a highly inter-connected node in these networks, its role in learning and controlling reaches remains underexplored. We report dissociable roles of two thalamic forelimb circuits coursing through parafascicular (Pf) and ventroanterior/ventrolateral (VAL) nuclei in refining reaches to a spatial target. Using 2-photon calcium imaging as mice learn directional reaches, we observe high reach-related activity from both circuits early in learning, which decreases with learning. Pf activity encodes reach direction early in learning, more so than VAL. Consistently, bilateral lesions of Pf before training impairs refinement of reach direction. Pre-training lesions of VAL does not affect reach direction, but increases reach speed and target overshoot. Lesions of either nucleus after training does not affect the execution of learned reaches. These findings reveal different thalamic circuits governing distinct aspects of learned reaches.

Skillfully reaching to targets in space is an important component of many actions, such as bringing a paint brush to a canvas or pen to paper. Reaches are learned and controlled through distributed motor networks, encompassing circuits such as the motor cortex[1–5], basal ganglia[6,7], cerebellum[8–11], and brainstem[12–14]. The basal ganglia and cerebellar circuits have been seen as having complementary roles in motor learning and execution, with the basal ganglia more involved in control and reinforcement of specific movements[15–19], and the cerebellum providing sensory feedback during ongoing actions to regulate the speed, smoothness, and endpoint precision of reaching actions[10,11,20].

There is clear evidence that the thalamus, which contains nuclei that receive inputs and send outputs to different motor centers, is a critical node in movement and motor learning[21,22]. The thalamic parafascicular (Pf) and the ventroanterior/ventrolateral (VAL) nuclei in particular, are situated at the convergence of important motor centers. Pf receives inputs from basal ganglia output nuclei[23–25], the superior colliculus[26,27], the cerebral cortex[21], and sparse collaterals from deep cerebellar nuclei (DCN)[28]. VAL receives inhibitory inputs from basal ganglia output nuclei as well[25,29,30], and receives dense

excitatory afferents from DCN[31,32] and motor cortex[33–35]. Pf and VAL's outputs differ significantly. Pf's glutamatergic outputs comprise the main sub-cortical input to the striatum, which are somatotopically organized[21] and almost equal in amount to the motor cortex inputs to striatum[36,37]. Pf outputs also target the subthalamic nucleus (STN)[38], another important movement center, and form reciprocal cortico-thalamic loops with limbic, associative, and somatosensory circuits[21,27]. VAL's outputs, on the other hand, primarily target layer 1, layer 3, and layer 5 of premotor and motor cortex, but not the striatum[29,39,40]. Pf and VAL's divergent anatomical footholds in these distributed motor networks suggest that they may underlie different aspects of learning and movement. Indeed, both Pf and VAL have been reported to have roles in learning and movement in health and disease[22,38,41,42]. However, it is unknown how their roles evolve with learning of a skilled action and what specific components of move-ments they encode and control.

Here, we show how Pf and VAL contribute to the control and refinement of reaches directed towards spatial targets, using a head-fixed joystick task that we developed[43]. We used 2-photon calcium imaging of glutamatergic neural populations in the forelimb regions of

[1]Department of Neuroscience, Zuckerman Mind Brain Behavior Institute, Columbia University, New York, NY, USA. [2]Institute of Neuroscience, University of Oregon, Eugene, OR, USA. [3]Allen Institute, Seattle, WA, USA. [4]These authors contributed equally: Leslie J. Sibener, Alice C. Mosberger. ✉e-mail: ljs2203@columbia.edu; rui.costa@alleninstitute.org

Pf and VAL across time, and found a large fraction of neurons with reach-related modulation in both areas. This reach-related activity decreased as learning progressed. By tracking the same neurons over time, we uncovered that population dynamics in Pf changed between early and late learning, but were stable after learning. In contrast, VAL's dynamics continuously shifted regardless of learning stage. Importantly, we show that neural activity in Pf specifically encodes the direction of reaches early in learning, but not once reaches are refined, and does so more than VAL. In accordance, pre-learning genetically targeted lesions revealed that glutamatergic neurons in Pf were required to refine reach direction during learning. VAL lesions, on the other hand, did not impair the refinement of reach direction, but resulted in target overshoot and faster movements. Post-learning lesions of either area did not affect the performance of learned reaches. These results demonstrate that Pf and VAL are critical nodes for motor behavior with distinct roles in learning forelimb reaching

movements, and endorses the view that the thalamus contains separable movement governing circuits.

## Results

### Mice explore and refine directional reaches to a spatial target

To study the roles of Pf and VAL in the learning and refinement of target reaches, we trained mice (n = 10) in our Spatial Target Task (STT)[43] while we imaged their thalamic neural activity (Fig. 1a, b). Head-fixed mice explored a 2-dimensional workspace by moving a joystick with their right forelimb from a set start location. Movements were rewarded when attempts entered a circular spatial target (hits), or aborted when the animal let go of the joystick or 7.5 s elapsed (misses) (Fig. 1c). During pre-training, mice were rewarded for movements in the general forward direction, and mouse-specific targets 40° to the left and right of the mean direction were defined on the last day of pre-training (Fig. S1a/b). Mice were then trained in two behavioral blocks

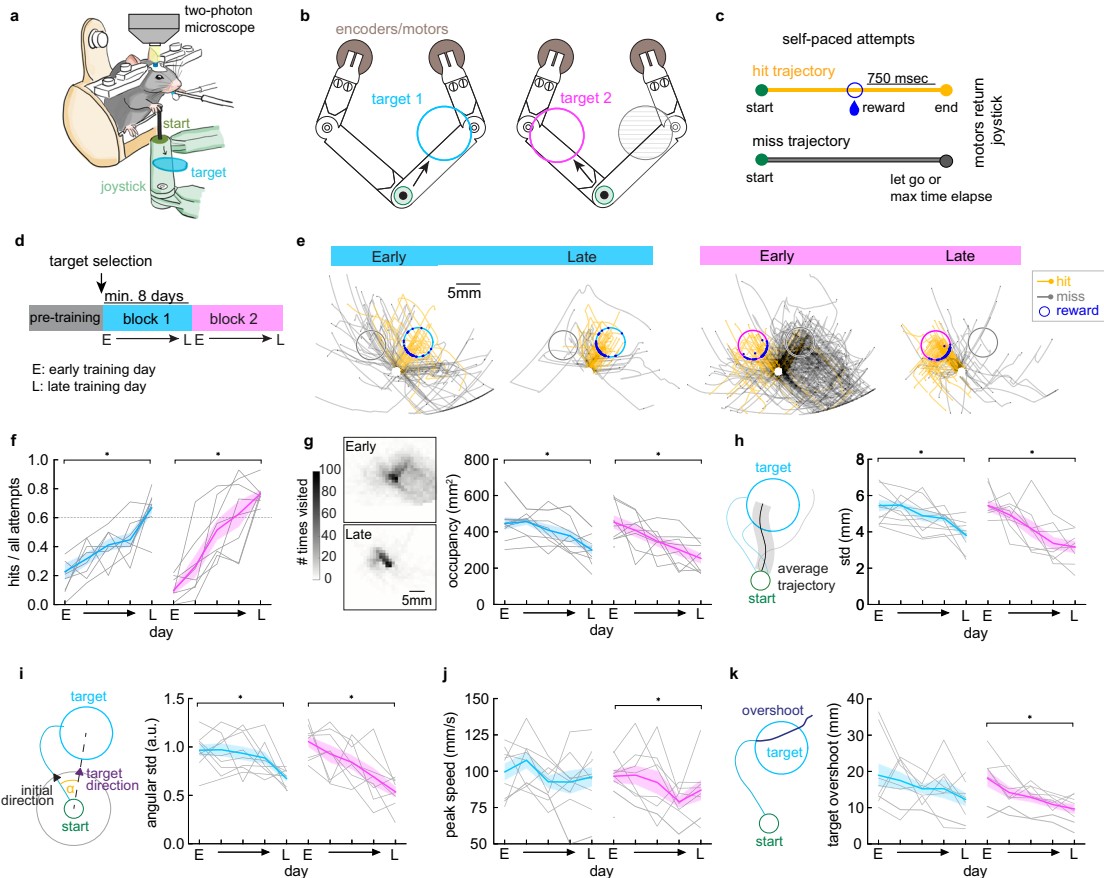

**Fig. 1 | Mice explore and refine directional spatial target reaches. a** Schematic of a head-fixed mouse performing the Spatial Target Task (STT) during imaging. **b** Left: top view schematic of the SCARA joystick. Start position (green circle) and example position of target 1 shown. Right: same as left, example of target 2 and previous target 1 (gray circle). **c** Classification of hit and miss trajectories from self-paced joystick movements. **d** Schematic showing training protocol. **e** Left: example trajectories on early and late days of block 1 with rewarded target (blue circle) and non-rewarding target (gray circle). Right: same for block 2 with rewarded target (magenta circle) and non-rewarding target (gray circle). Hit trajectories (yellow) and miss trajectories (gray) are plotted. Point of target entry and reward delivery is shown (blue dot). **f** Hit ratio on 5 selected days per block from early training day (E) to late training day (L) and 3 equidistant days in between. Mixed-effects model, day in block: $F_{(3,25)} = 40$, *$p < 0.0001$, day x block: $F_{(3,14)} = 4$, *$p < 0.05$. Šídák's multiple comparisons test: E vs L block 1, *$p < 0.005$; E vs L block 2, *$p < 0.0001$. **g** Left: representative heatmap of workspace occupancy of all trajectories of an early and late session. Right: occupancy of workspace (mm²). Mixed-effects model, day in block: $F_{(3,23)} = 13$, *$p < 0.0001$, mouse: $F_{(1,9)} = 6$, *$p < 0.05$. Šídák's multiple

comparisons test: E vs L block 1, *$p < 0.05$; E vs L block 2, *$p < 0.005$. **h** Variability of mean trajectory from all movements. Mixed-effects model, day in block: $F_{(2,20)} = 19$, *$p < 0.0001$, mouse: $F_{(1,9)} = 17$, *$p < 0.01$. Šídák's multiple comparisons test: E vs L block 1, *$p < 0.005$; E vs L block 2, *$p < 0.005$. **i** Variability of initial direction of movement. Mixed-effects model, day in block: $F_{(3,22)} = 8$, *$p < 0.01$, mouse: $F_{(1,9)} = 7$, *$p < 0.5$. Šídák's multiple comparisons test: E vs L block 1, *$p < 0.05$; E vs L block 2, *$p < 0.005$. **j** Peak speed of trajectories. Mixed-effects model, day in block: $F_{(4, 36)} = 4$, *$p < 0.05$, mouse: $F_{(1,9)} = 8$, *$p < 0.05$. Šídák's multiple comparisons test: E vs L block 1, $p > 0.05$; E vs L block 2, *$p < 0.05$. **k** Average target overshoot from target entry for all hit trajectories. Mixed-effects model, $F_{(3,25)} = 8$, *$p < 0.001$, mouse: $F_{(1,9)} = 9$, *$p < 0.05$. Šídák's multiple comparisons test: E vs L block 1, $p > 0.05$; E vs L block 2, *$p < 0.05$. **f–k** Mean ± SEM is shown in thick color lines and shaded bounds (block 1: blue, block 2: magenta), single animals shown in gray lines (n = 10 mice). Asterisks indicate Šídák's multiple comparisons test for E vs L day in block. See also Supplementary Fig. 1. Source data are provided as a Source Data file.

with one active target per block, resulting in different directional reaches learned in each block (Fig. 1d/e). The block switch and active target change to target 2 occurred the day after the animal reached a high-performance criterion on target 1 (>60% average hit ratio over three consecutive days), or a maximum number of days (53 days) had elapsed (Fig. S1c). In order to compare results across animals with different block lengths, we selected five equidistant days per block for each animal (see "Methods").

Mice gradually increased their performance to reach high hit ratios late in the block (Fig. 1f). As animals learned to hit the rewarded target area, their reach trajectories occupied less of the workspace (Fig. 1g), and the variability of their mean trajectory significantly decreased (Fig. 1h). To investigate if mice refined the initial direction of reaching movements, we measured the variability of the initial movement vector angle using the angular std[44], and found that it significantly decreased within each block (Fig. 1i). In addition to learning to reach in the correct direction, we find an overall decrease in the peak speed of reaches across animals (Fig. 1j), suggesting that they slowed down to accurately hit the target. In line with this finding, we also found that mice significantly reduced the target overshoot pathlength with learning (Fig. 1k).

These findings suggest that as mice learned to reach the rewarded targets, they refined the initial direction of movement and reduced peak speed, while increasing targeting precision. Furthermore, on the first day of block 2 when the new target was rewarded, even though mice initially perseverated with movements to the previously rewarded target (Fig. S1d), they promptly increased the variability of their trajectory and initial reach direction (Fig. 1h, i). This change in behavior indicates an increase in exploration to discover the new target direction.

## $Pf_{FL}$ and $VAL_{FL}$ neural populations are most active during reaches early in learning

To investigate the role of thalamic nuclei in the learning and performance of forelimb reaches, we first mapped the forelimb-related thalamic areas. Previous studies have detailed the somatotopy of primary motor cortex (M1) and identified a consistent caudal forelimb area (CFA) and rostral forelimb area (RFA)[45]. Similarly, the dorsal striatum has been shown to contain distinct areas that receive input from sensorimotor cortex according to their somatotopy[46,47], which identified a dorsolateral striatal forelimb-related area ($DLS_{FL}$). Strongly projecting to these major sensorimotor regions, Pf and VAL also have topographic organizations[21,35,48]. To specifically map the forelimb-related thalamus we injected the retrograde tracer cholera toxin subunit B (CTB) into $DLS_{FL}$ and the CFA of M1 in wildtype mice (Fig. S2a/b and e/f). We serially processed and reconstructed the brains using the BrainJ[49] pipeline and aligned them to the updated Allen Brain Atlas (ABA)[50]. This atlas alignment showed that Pf and VAL were the most strongly labeled thalamic nuclei when CTB was injected into $DLS_{FL}$ and CFA, respectively (Fig. S2c, g). Using these data, we delineated the forelimb-related thalamic areas ($Pf_{FL}$ and $VAL_{FL}$) by reconstructing the regions with detected cell bodies and integrated those volumes as new somatotopic areas in the ABA (Fig. S2d/h).

To measure the activity of glutamatergic cells[51,52] in $Pf_{FL}$ and $VAL_{FL}$ during learning and performance of targeted reaches, we injected AAV5 expressing GCamp6f under a CamKII promoter into these respective areas (Fig. S3a), followed by implantation of a gradient index (GRIN) lens, and recorded single cell calcium activity using a 2-photon microscope throughout training of the STT in BL6 mice (Fig. 2a). Cells were segmented and calcium transients extracted using Suite2p[53] (Fig. 2b). Individual cell traces were detrended and z-scored across daily sessions for all further analysis (Fig. 2c). Imaging yielded a stable number of cells across several weeks of behavioral training (Pf: average neurons / animal / day 34 ± 3, VAL: average neurons / animal / day 33 ± 4) (Fig. S3c/d). In order to investigate whether neural activity

in $Pf_{FL}$ and $VAL_{FL}$ was modulated during reaching movements, we aligned the neural activity of all cells to movement initiation (leaving start position) on multiple days of block 1 and 2 (Fig. 2c) and averaged their activity across trials (Fig. 2d). In both thalamic areas, cells displayed diverse activity modulation, including increasing, decreasing, or maintaining their average activity around the movement initiation. To test whether their activity was most modulated before, during, or after the target reach, we compared the ratio of significantly modulated cells during three time windows around movement on the early day of block 1: pre-movement, -300-0 ms before movement initiation; during movement, 0–300 ms after leaving the start position; and after target hit, 0–300 ms following target hit (Fig. 2e). Cells were classified as significantly modulated if their trial-averaged activity within the defined time window was above or below 3x the SD of that cell's baseline. For $Pf_{FL}$ and $VAL_{FL}$ animals, a significantly larger proportion of cells were modulated during the reach than before reach start, or after target hit (Fig. 2f).

We wondered whether this reach-related activity changed over the course of learning in either thalamic area. To this end, we created trial-averaged, population-averaged activity traces for three selected days of each block across all animals. In both blocks, we found that the overall activity in $Pf_{FL}$ was high early in the block, but significantly decreased with learning as the reaches refined (Fig. 2g, h, top). Activity in $VAL_{FL}$ also decreased with learning in the first block, but did not return to a high magnitude early in block 2 and was low throughout the second block (Fig. 2g, h, bottom). This decrease in average activity over learning could be explained by a change in the relative numbers of positively and negatively modulated cells. By classifying cells into positively, negatively, or not significantly modulated during movement (Fig. 2i and Fig. S4a/b), we found that the relative numbers of modulated cells in $Pf_{FL}$ changed from early to late days in block 1 (Fig. 2i and Fig. S4e) and that the amplitude of positively modulated cell responses was highest early in learning (Fig. S4c, left). This indicates that more cells in $Pf_{FL}$ were positively modulated and their response magnitudes were larger early in training. $VAL_{FL}$ modulated cell response distributions remained relatively consistent throughout training (Fig. S4f). Surprisingly, in $VAL_{FL}$ we found that the amplitudes of positively modulated cells decreased over learning, while the negatively modulated cells increased their magnitudes over time (Fig. S4d). By classifying cell response, we also revealed that there was a larger ratio of negatively modulated cells in $VAL_{FL}$ compared to $Pf_{FL}$ across all analyzed days (Fig. S4e/f), explaining the overall lower average activity in $VAL_{FL}$. Taken together, $Pf_{FL}$ and $VAL_{FL}$ cells are mostly active during the execution of forelimb reaches in our task, and that the magnitude of $Pf_{FL}$ population activity decreased with learning as fewer cells were positively modulated with smaller amplitudes during the reach as training progressed. In contrast, overall $VAL_{FL}$ population activity decreased across training, which was driven by an increase in amplitude of negatively modulated cells.

## Modulation of neural activity during movement changes across learning

We next asked whether the observed changes in population activity were due to different cells being engaged, or the same cells modulating their activity differently across stages of learning. To investigate this, we repeated the previous analysis using matched cells from two early days (early 1 and early 2) and two late days (late 1 and late 2) per block, as well as cells that were matched across the late day of block 1 and the early day of block 2 (which is the first day of block 2) (Fig. 3a, b). To match cells across days, we used non-rigid motion correction to align the imaging fields of view from the selected days and matched the segmented cells (Fig. S5). We found that the reach-related population activity of matched cells in $Pf_{FL}$ was significantly higher on the first early day than the second early day for both block 1

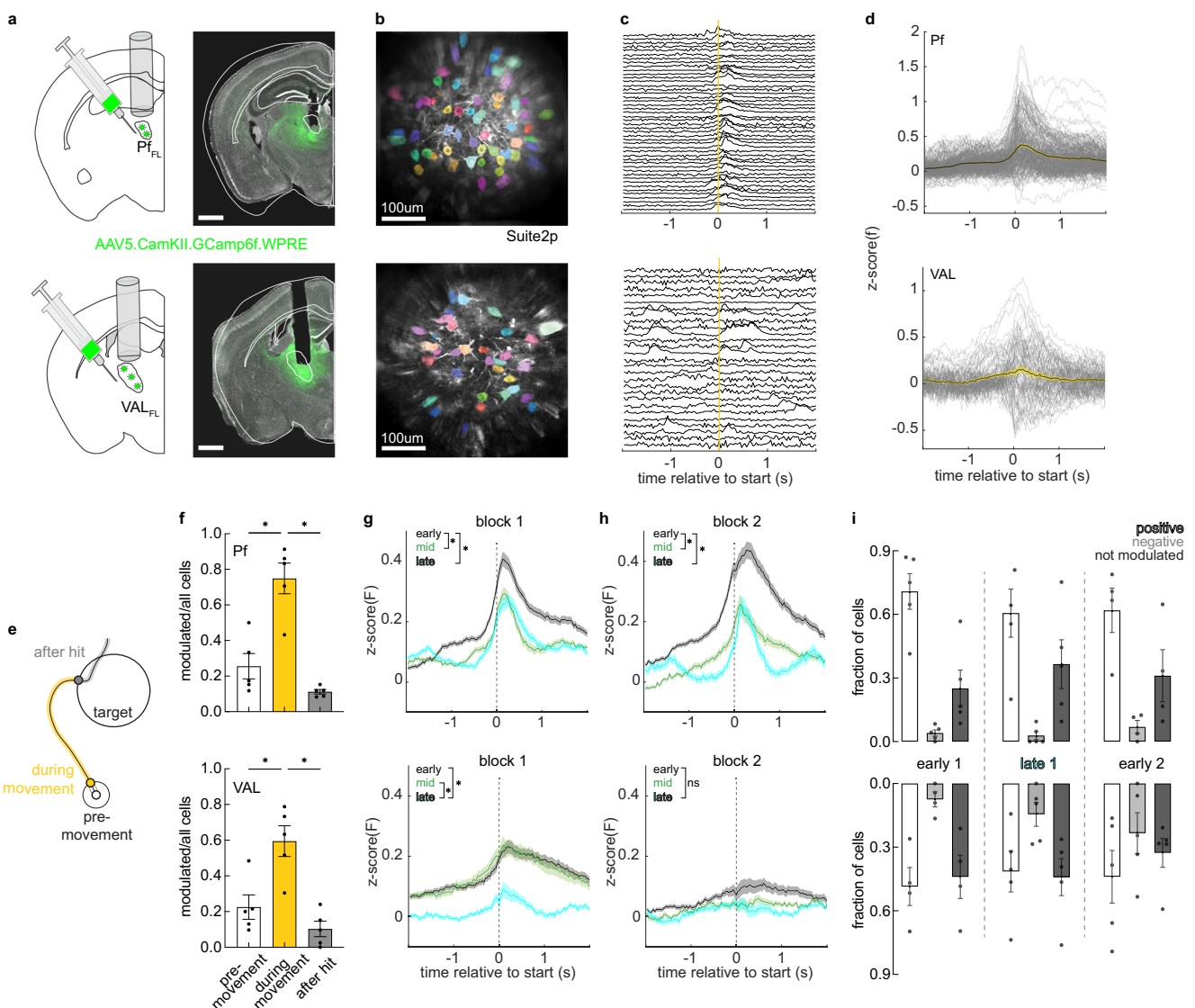

**Fig. 2 | Pf_FL and VAL_FL neural populations are most active during reaches early in learning. a** Top: schematic and representative histology of viral targeting and GRIN lens implantation for calcium imaging in Pf_FL. Bottom: same for VAL_FL. Scale represents 1 mm. **b** Top: example average fluorescence of the imaging field of view with detected ROI masks overlayed for Pf_FL. Bottom: same for VAL_FL. Scale represents 100 μm. **c** Top: example detrended and z-scored calcium traces of cells aligned to a reach start (yellow line) for Pf_FL. Bottom: same for VAL_FL. **d** Top: trial-averaged z-scored fluorescence aligned to reach start for Pf_FL cells on example day. Mean ± SEM (black trace and yellow shaded bounds). Individual cell trial-averages (gray); n = 213 neurons, 5 mice. Bottom: same for VAL_FL; n = 189 neurons, 5 mice. **e** Schematic of the three time windows of a reach; pre-movement (white), during movement (yellow), after hit (gray). **f** Top: proportion of significantly modulated cells on early day of block 1 during the time windows depicted in (**e**) for Pf_FL, n = 5 mice. Pf_FL: Repeated measures one-way ANOVA, F(2,7) = 31, *p = 0.0003; Šídák's multiple comparisons: pre-movement vs during movement: *p = 0.008, during movement vs after hit: *p = 0.006. Bottom: same for VAL_FL, n = 5 mice. Repeated measures one-way ANOVA: F(1,6) = 15, *p = 0.007; Šídák's multiple comparisons:

pre-movement vs during movement: *p = 0.008, during movement vs after hit: *p = 0.03. Data shown as mean ± SEM. **g** Top: trial-average z-scored fluorescence of all cells on 3 sessions from block 1; early (black), middle (green), late (cyan) for Pf_FL. Kruskal-Wallis test, population activity at start: *p < 0.0001; Dunn's multiple comparisons test, early vs mid: *p < 0.005, early vs late: *p < 0.0001, mid vs late: p > 0.05. Bottom: same as top for VAL_FL data. Kruskal-Wallis test, population activity at start: *p < 0.001; Dunn's multiple comparisons test, early vs mid: *p < 0.005, early vs late: *p < 0.005, mid vs late: *p < 0.0001. Data shown as mean ± SEM. **h** Same as (**g**) for block 2; Top: Pf_FL Kruskal-Wallis test, population activity at start: *p < 0.0001; Dunn's multiple comparisons test, early vs mid: *p < 0.0001, early vs late: *p < 0.0001, mid vs late: p > 0.05. Bottom: same as top for VAL_FL data. Kruskal-Wallis test, population activity at start: p > 0.05. Data shown as mean ± SEM. **i** Fraction of cells that are positively (white bar), negatively (gray bar), or not significantly (dark gray bar) modulated during movement for Pf (top, n = 5 mice) and VAL (bottom, n = 5 mice). Data shown from early block 1 (left), late block 1 (middle), and early block 2 (right). Data shown as mean ± SEM. See also Supplementary Fig. 2–4. Source data are provided as a Source Data file.

and 2 (Fig. 3a, early 1 vs early 2), but there was no difference in match cell population activity for the two late days in either block (Fig. 3a, late 1 vs late 2). Additionally, Pf_FL matched cell population activity also increased with the block switch (Fig. 3a, block switch). This shows that even when tracking matched cell populations, Pf_FL reach-related activity is highest early in training, and decreases over both blocks. In contrast, VAL_FL matched cell activity only changed between two early

days in block 1 and did not change across the block switch or early in block 2 (Fig. 3b).

We next determined which cells were positively, negatively, and non-modulated and compared the distribution of each modulation type over matched days (Fig. S6a, b). For Pf_FL we found that the distribution of match cell population responses shifted significantly early in block 1 and block 2, but not for any other matched days (Fig. 3c).

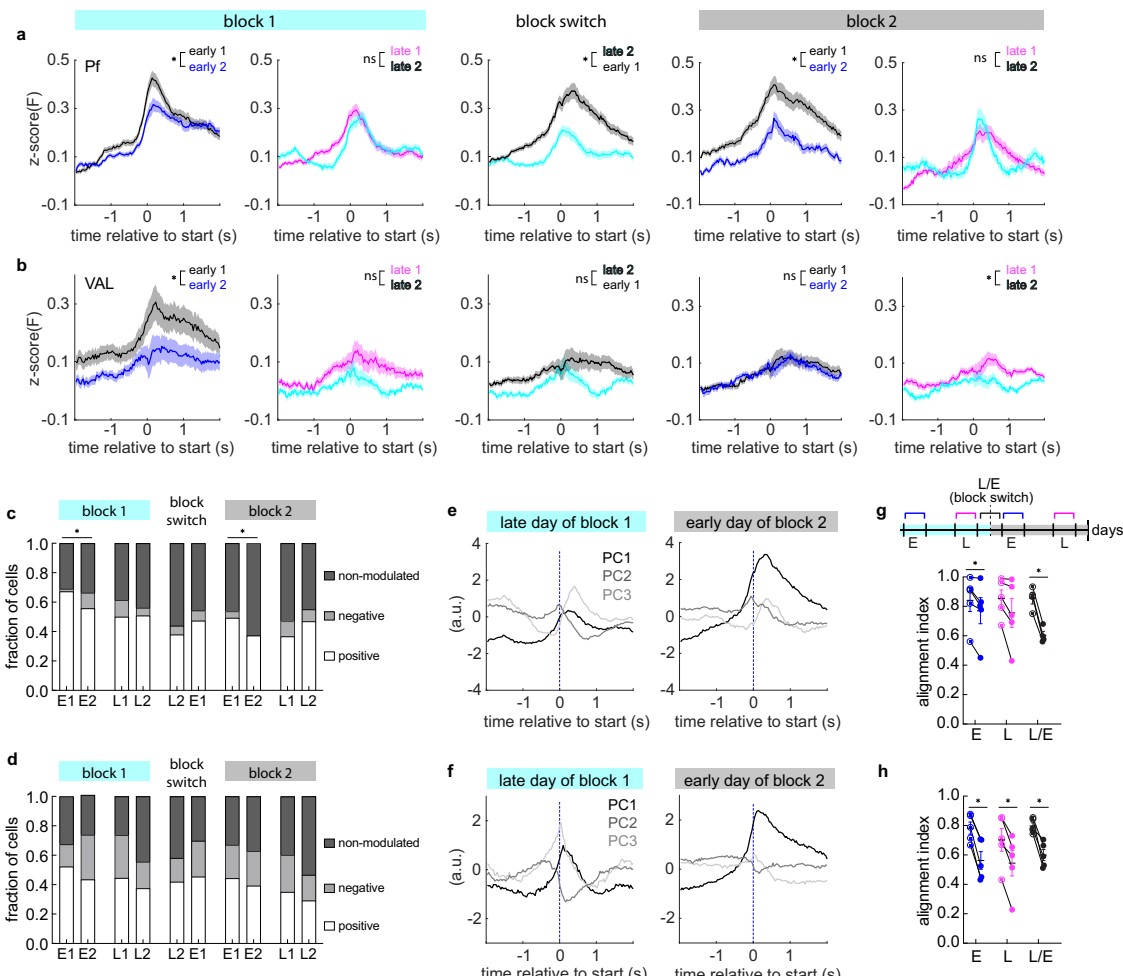

**Fig. 3 | Low dimensional population activity alignment changes between early and late learning. a** $Pf_{FL}$ trial-averaged fluorescence aligned to movement start for matched cells on two early days of block 1 (n = 113 cells, 5 mice), two late days of block 1 (n = 114 cells, 5 mice), two days around block switch (n = 116 cells, 5 mice), two early days of block 2 (n = 67 cells, 4 mice), and two late days of block 2 (n = 49 cells, 4 mice). Wilcoxon matched-pairs signed rank test, two-sided: early 1–2 (block 1) *p < 0.0001, late 1–2 (block 1) p > 0.05, late 2 (block 1) – early 1 (block 2) *p < 0.0001, early 1–2 (block 2). *p < 0.0001, late 1–2 (block 2) p > 0.05. Data shown as mean ± SEM. **b** Same as in (**a**) for $VAL_{FL}$. Two early days of block 1 (n = 46 cells, 4 mice), two late days of block 1 (n = 72 cells, 5 mice), two days around block switch (n = 86, 5 mice), two early days of block 2 (n = 97 cells, 5 mice), and two late days of block 2 (n = 103 cells, 5 mice). Wilcoxon matched-pairs signed rank test, two-sided: early 1–2 (block 1) *p = 0.0008, late 1–2 (block 1) p > 0.05, late 2 (block 1) – early 1 (block 2) p > 0.05, early 1–2 (block 2). p > 0.05, late 1–2 (block 2) *p = 0.01. **c** $Pf_{FL}$ population distribution of positively (white), negatively (gray), and non-modulated

(dark gray) cells on all matched day pairs. Fisher's exact test (performed on cell counts): E1-E2 (block 1) *p = 0.013, E1-E2 (block 2) *p = 0.045. All other matched day pairs' Fisher's exact tests: p > 0.05. **d** Same as in (**c**) for $VAL_{FL}$. Fisher's exact test for all matched day pairs: p > 0.05. **e** Top 3 PCs of neural activity of $Pf_{FL}$ matched cell populations around movement start for late day of block 1 and early day for block 2. **f** Same as (**d**), for $VAL_{FL}$ matched neuronal populations. **g** For $Pf_{FL}$, average alignment of subspaces occupied within (open circle with dot) and across (filled circle) days with matched cells from two early days (E) in each block (black, average over block 1 and 2), two late days (L) in each block (magenta, average over block 1 and 2), and the block switch day from late block 1 to early block 2 (L/E, blue). For $Pf_{FL}$ (n = 5 mice), two-tailed paired t test within vs across; E, *p = 0.026; L, p > 0.05; L, E, *p = 0.0015. center and error bars indicate mean ± SEM. **h** Same as (**g**), for $VAL_{FL}$ (n = 5 mice) matched cell populations. Two-tailed paired t test within vs across; E, *p = 0.0008; L, *p = 0.0037; L/E, *p = 0.0073. Error bars shown as mean ± SEM. See also Supplementary Fig. 6–8. Source data are provided as a Source Data file.

---

By comparing the trial average responses of positively modulated cells, we found that the magnitude of these cells increased at the switch from block 1 to 2, and rapidly decreased in early block 2 training (Fig. S6c). These data indicate that the increased population activity on the early day of block 2 was driven both by a subpopulation of newly upmodulated cells and increased magnitude of responses.

In contrast, $VAL_{FL}$ matched population response distributions were not different for any matched day pairs (Fig. 3d). Negatively modulated cells increased their response magnitude across early days of block 1 (Fig. S6f), which explains the decrease in overall population response (Fig. 3b), but there were no other changes in magnitude of negatively or positively modulated cells across either block (Fig. S6e, f). These results suggest that the population changes seen in $VAL_{FL}$

activity were primarily driven by an increase in response of negatively modulated cells over learning.

## The alignment of population dynamics changes between early and late learning

After observing changes at the single cell level in $Pf_{FL}$ and $VAL_{FL}$, we wanted to characterize their populations' low-dimensional neural dynamics, and ask whether those dynamics also shifted over time. Previous work has documented such low-dimensional dynamics during reaching movements in other motor centers, such as M1[4]. To find the dominant patterns of covariance across neurons we applied principal component analysis (PCA) on the trial-averaged activity of matched cells around the time of movement start. First, we visualized

the movement-locked population activity of the top 3 PCs for two days around block switch (Fig. 3e, f), and two late days in block 1 (Fig. S7a-b) by running a single PCA on the concatenated time windows from matched days. Movement-locked population activity on days spanning the block switch occupied different dimensions within the space spanned by the first 3 PCs for Pf$_{FL}$ (Fig. 3e) and VAL$_{FL}$ (Fig. 3f), with large changes along PC1 on the early day of block 2, indicating the dominant dimensions of covariance changed at the block transition. In comparison, the activity projected onto the top 3 PCs appeared to traverse through PC space more similarly between the two days late in block 1 (Fig. S7a-b). Across all matched days, the top three PCs explained over 80% of variance in Pf$_{FL}$ (82% ± 6%) and over 75% for VAL$_{FL}$ (77% ± 7%) (Fig. S7c-d), confirming that the dominant neural dynamics were captured in only a few dimensions.

To test whether the dominant dimensions of neural covariance changed when the new target was rewarded, we calculated the PCs for each animal and day individually. We then compared the PC alignment for each animal within the same day, to check for changes of low dimensional activity over a single session. Next, we did the same across different days, to test if there were changes in alignment between days and stages of learning (see Methods). For Pf$_{FL}$, we found that the dimensions containing the most variance were significantly different between two early days (Fig. 3g, in blue), and the late day of training of block 1 and the first day of block 2 (Fig. 3g, in black). This indicates a change in the main direction of variance between early and late learning. However, the alignment across late days was not significantly different for Pf$_{FL}$ (Fig. 3g, in magenta), suggesting that the dominant dimensions of neural covariance from day to day stabilized or crystalized after learning. In contrast, the alignment of dimensions for VAL$_{FL}$ was significantly different across all matched-day comparisons (Fig. 3h), suggesting that VAL$_{FL}$'s neural activity did not stabilize with learning. This is in stark contrast to Pf$_{FL}$, which did not change the dominant dimensions of covariance late in learning. Importantly, the alignment across sessions for Pf$_{FL}$ and VAL$_{FL}$ was reduced when neuron identity was shuffled (paired t-test for all matched days, across vs across-shuffle: *p < 0.05 for all comparisons), emphasizing that dimensions are more aligned using matched cells across days than if random pairs were used, and that cellular identity across days is a key component of finding meaningful low-dimensional representations across learning. In summary, by tracking the same neurons during training, we found that neural population dynamics of Pf$_{FL}$ occupied different dimensions during early vs. late learning of directional reaches, but stabilized late in learning, which suggests a role for Pf$_{FL}$ in the learning process itself. In contrast, VAL$_{FL}$ occupied significantly different dimensions each day, even after learning.

## Pf$_{FL}$ neural activity specifically predicts reach direction early in learning

We next investigated whether neural activity encoded specific aspects of the reaching movements and how this encoding changed with learning. We first analyzed if single cells were not only tuned to movement onset but also to specific reach directions. We split individual trials into 60 degree bins and compared the peak trial-average activity for different directions. A cell was considered tuned to a specific direction if an overall ANOVA across all directions was significant and the highest peak activity was significantly different than the peak activity of at least one other direction (multiple comparison) (Fig. S8a, example cells). The proportion of cells with direction tuning was highest early in learning, and Pf$_{FL}$ had significantly more direction tuned cells than VAL$_{FL}$ early in learning (Fig. S8b-c).

We next analyzed whether the initial reach direction could be decoded from the neural activity on a single-trial basis. We

trained Ridge regression models to predict the sine and cosine of the initial vector angle from the neural activity (0–167 msec after start) on held out test trials (10 models with different 70%/30% train/test splits per session), and the average coefficient of determination ($R^2$) between the true direction and the predicted direction across all splits (Fig. 4a–d) was used to compare decoding accuracy between animals and days. A model that always predicts the mean of the distribution would result in a coefficient of determination of zero, and a model performing worse can yield a value less than zero. The coefficient of determination is bound to correlate with the variance of the dependent variable (initial direction), which is inherently higher on early days than late days in our task (Fig. S8d). To compare model accuracy between early and late days, we used subsampled trials to a matched level of directional variance (Fig. S9, see Methods) and used these variance-matched trials for all models. We found that initial direction was better decoded from Pf$_{FL}$ neural activity early in learning than late (Fig. 4e). Furthermore, Pf$_{FL}$ neural activity predicted the direction of movement significantly better than VAL$_{FL}$ neural activity (Fig. 4e). Models in which the trial identity was shuffled in the train dataset performed poorly on all days for Pf$_{FL}$ and VAL$_{FL}$ (Fig. 4e and Fig. S8d). We further found that Pf$_{FL}$ neural activity better encoded reach direction during movement than before movement onset (167–0 msec before start) (Fig. 4f). This was not the case for VAL$_{FL}$ neural activity (Fig. 4g). These findings further support a specific role for Pf in conveying the ongoing reach direction to striatum, but not necessarily planning or generating the reach direction itself.

We next investigated if other movement aspects, such as the maximum speed or maximum distance from the start position that is reached in each trial, can be similarly predicted from either Pf$_{FL}$ or VAL$_{FL}$ neural activity. We trained multivariate Ridge regression models to predict the sine/cosine of the initial direction, as well as the maximum speed and maximum distance for each trial (Fig. S8e, Full Model). We then trained additional models where one of the predicted variables (direction, speed, or distance) was removed (Fig. S8f–h) to determine the relative contribution of each variable to model performance. Comparing the accuracy of these different models on early days of the block revealed that for both Pf$_{FL}$ and VAL$_{FL}$, initial direction was the best decoded variable (Fig. 4h). The model predicting maximum speed and distance showed significantly lower accuracy than models that also used direction information. Furthermore, for Pf$_{FL}$ animals, the Full Model performed significantly worse than the model using direction information alone. Models trained to only predict the maximum speed (Fig. S8i) or maximum distance (Fig. S8j) on separately variance-matched trials, had low decoding accuracy for all animals and sessions. Taken together, these results show that Pf's movement-related activity contains information specifically about the initial direction of the reaching movement during the movement itself, and that information decreases with learning. Additionally, this direction information is significantly less encoded in VAL.

## Ablation of Pf$_{FL}$, but not VAL$_{FL}$, cells before learning impairs the refinement of reach direction

Based on the results above, we hypothesize that Pf$_{FL}$'s reach-related activity early in training, which predicted reach direction (Figs. 2, 4), is needed to learn and refine the direction of target reaches. To test this, we performed pre-learning lesions of Pf$_{FL}$ and VAL$_{FL}$. We selectively ablated glutamatergic cells by bilaterally injecting AAV5-flex-taCasp3-TEVp in the Pf$_{FL}$ or VAL$_{FL}$ of VGlut2-Cre mice and WT littermates (Fig. 5a) (Pf$_{pre-lesion}$, n = 8; Pf$_{pre-control}$, n = 10; VAL$_{pre-lesion}$, n = 7; VAL$_{pre-control}$, n = 10), which induced cell death through apoptosis[54]. Lesions were confirmed by histological staining against the neuronal cell marker NeuroTrace. Lesioned animals were left with large areas

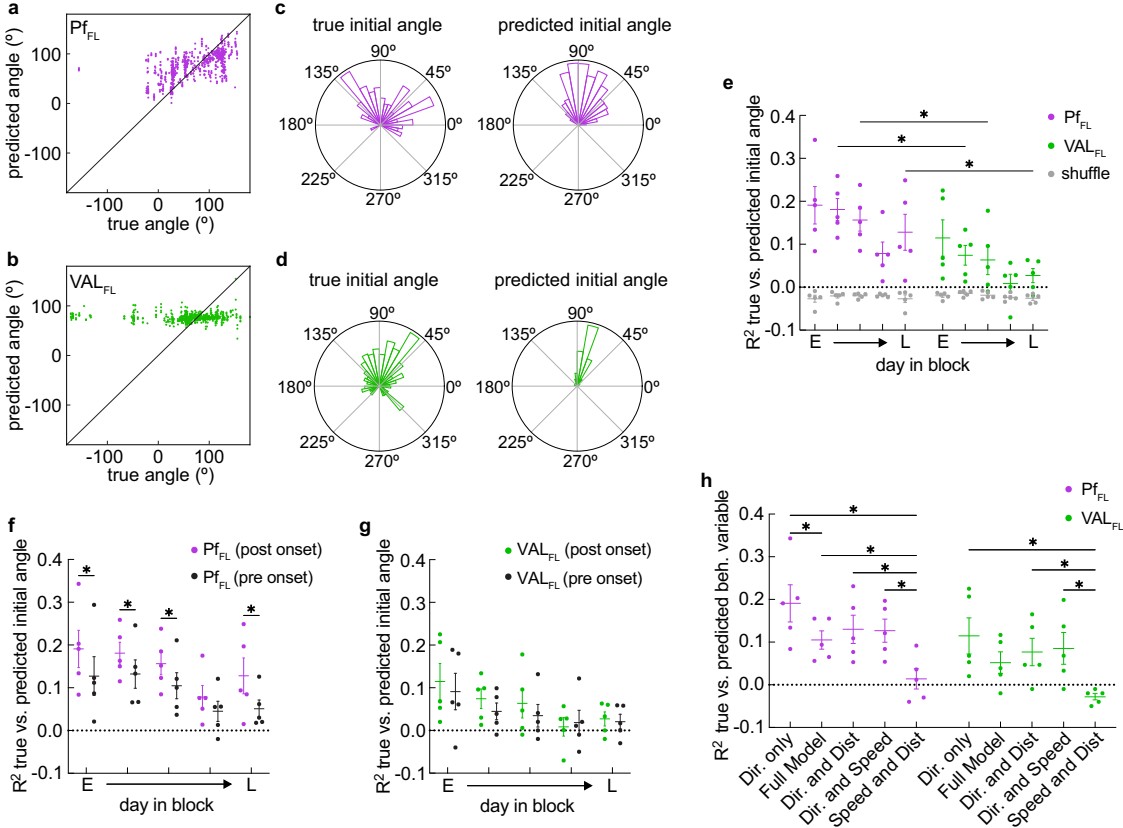

**Fig. 4 | Pf_FL neural activity encodes reach direction early, but not late, in learning. a** Predicted initial vector direction of combined held-out test data from 10 models trained on Pf_FL neural activity using different train/test splits, plotted against the corresponding true initial vector direction. Models trained on an early session in block 1. **b** Same as in (**a**), for VAL_FL animal. **c** Circular histogram showing the true (left) and predicted (right) initial vector directions from a representative Pf_FL animal data in (**a**). **d** Circular histogram showing the true (left) and predicted (right) initial vector directions from a representative VAL_FL animal data in (**b**). **e** Averaged coefficients of determination ($R^2$s) of model predictions for Pf_FL (purple), VAL_FL (green), and models trained on shuffled vector data (gray) for 5 days of training in each block (averaged over two blocks). 2-way repeated measures ANOVA: day effect, $F(4,32) = 4$: *$p = 0.0115$, region effect, $F(1,8) = 17$: *$p = 0.0031$. Šídák's multiple comparisons test: Pf_FL early-mid vs VAL_FL early-mid, *$p = 0.021$. Pf_FL mid vs VAL_FL mid, *$p = 0.043$. Pf_FL late vs VAL_FL late, *$p = 0.029$. For all other day comparisons, $p > 0.05$. For shuffled data: 2-way repeated measures ANOVA, day and region effects, $p > 0.05$. **f** Averaged coefficients of determination ($R^2$s) of model predictions for Pf_FL using neural activity pre (black) and post (purple) onset of movement.

2-way repeated measures ANOVA: time from onset effect, $F(1,4) = 62$: *$p = 0.0014$. Šídák's multiple comparisons test: pre vs post (early), *$p = 0.0008$, pre vs post (early-mid), *$p = 0.0084$, pre vs post (mid), *$p = 0.0052$, pre vs post (mid-late), $p > 0.05$, pre vs post (late), *$p = 0.0001$. **g** Same as in (**f**), for VAL_FL neural data pre (black) and post (green) movement onset. 2-way repeated measures ANOVA: time from onset effect, $p > 0.05$. **h** Averaged coefficients of determination ($R^2$s) of multivariate model predictions for Pf_FL (purple), VAL_FL (green), early day block average. 2-way repeated measures ANOVA for Pf_FL: Model effect, $F(4,32) = 19$: *$p < 0.0001$. Šídák's multiple comparisons test for Pf_FL: direction vs full model, *$p = 0.0353$. Direction vs speed/distance, *$p < 0.0001$. Full model vs speed/distance, *$p = 0.0219$. Direction/distance vs speed/distance, *$p = 0.0018$. Direction/speed vs speed/distance, *$p = 0.0025$. For all other comparisons, $p > 0.05$. Šídák's multiple comparisons test for VAL_FL: Direction vs speed/distance, *$p = 0.0001$. Direction/distance vs speed/distance, *$p = 0.0056$. Direction/speed vs speed/distance, *$p = 0.0023$. For all other comparisons, $p > 0.05$. **e–g** Center and error bars indicate mean ± SEM. See also Supplementary Fig. 8, 9. Source data are provided as a Source Data file.

devoid of healthy NeuroTrace staining, with clear morphological differences to the WT control animals (Fig. 5b–e). Reconstruction of the 3D volumes of the lesions showed Pf and VAL were the primary targets of the lesions over neighboring thalamic nuclei (Fig. S10d–g), and that the percent of Pf and VAL lesioned did not correlate with overall performance (Fig. S10h/i).

Behavioral training began three weeks after AAV injection and all animals were trained for a maximum number of days in each block (Fig. 5a and Fig. S10a/b). Pf_pre-lesion mice showed a strong deficit in learning to hit the rewarded target area in both blocks compared to the Pf_pre-control group (Fig. 5f), even though the two groups were equally engaged in the task and performed the same number of attempts (Fig. S10c). VAL_pre-lesion mice, however, did not have such a learning deficit (Fig. 5g and Fig. S10b), and were also equally engaged in the task as their controls (Fig. S10c). These results suggest that Pf neural activity is needed to learn directional reaches.

Seeing that performance was affected, we wanted to know whether Pf_pre-lesion animals had a specific impairment in their ability to refine the direction of movements, potentially underlying their poor performance. We found that Pf_pre-lesion mice maintained high variability in the initial direction of their reaches throughout training, indicating that they were indeed unable to refine their reach direction (Fig. 5h). However, there was no effect on other movement aspects, such as speed (Fig. 5j) and target overshoot (Fig. 5l). In contrast, VAL_pre-lesion mice refined their initial reach direction with learning similar to their controls (Fig. 5i), allowing them to increase their hit ratio with training. Interestingly, we found that VAL_pre-lesion reaches moved with higher peak speeds (Fig. 5k), and had larger target overshoots (Fig. 5m). However, this effect on speed and targeting did not disrupt successful hitting of the target and overall performance.

These results show that both Pf and VAL lesions caused specific impairments and changes in spatial target reaches. In line with our

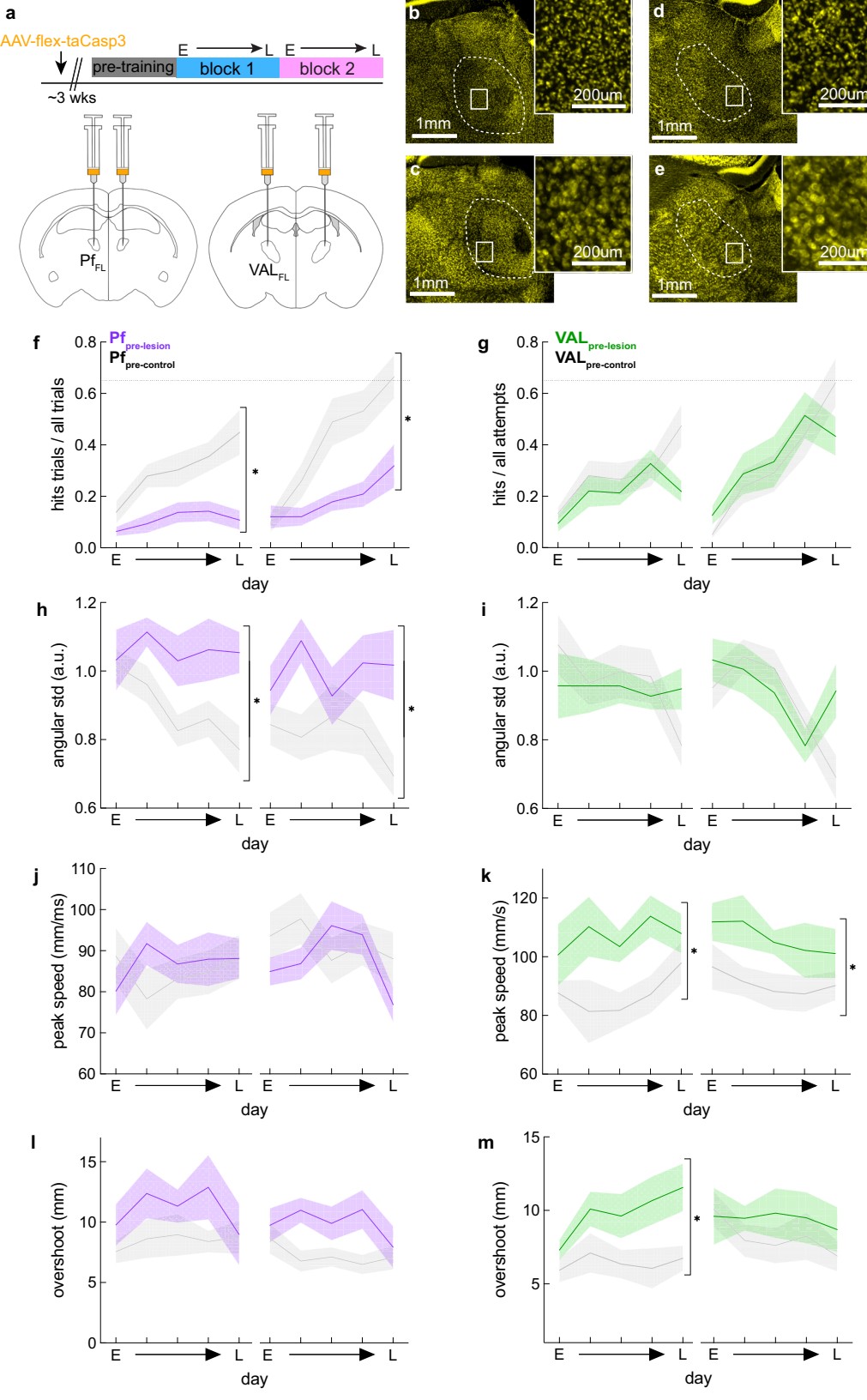

neural decoding results, lesioning Pf, but not VAL, led to an impairment in the refinement of the initial direction of reaches, resulting in a strong learning deficit. However, lesioning VAL resulted in animals moving faster and overshooting the target more than their controls. Taken together, these results reveal separable functional roles of these two thalamic nuclei in forelimb reaching movements.

## Pf and VAL are not required for the performance of learned directional reaches

Given our findings that Pf neural activity and direction prediction accuracy were strongest early in learning, but decreased or disappeared on late days with more refined movements, we hypothesized that while these thalamic areas are required early in learning, they are

**Fig. 5 | Ablation of Pf$_{FL}$, but not VAL$_{FL}$, cells before learning impairs the refinement of initial reach direction. a** Experimental design showing the timeline for surgeries and STT training, and coronal ABA sections showing the injection targets for Pf$_{FL}$ and VAL$_{FL}$ bilateral lesions. **b** Representative histology of Pf$_{pre-lesion}$ mouse, dashed-line indicating Pf. NeuroTrace (yellow), with high-magnification view from the inset in box. **c** Same as (**b**) for Pf$_{pre-control}$ mouse. **d, e** Same as (**b, c**), for VAL$_{pre-lesion}$ and VAL$_{pre-control}$ mice. Dashed-line indicating VAL. **f** Hit ratio on five selected days over two blocks; Early (E) day of block, Late (L) day of block, and three equidistant days in between. Pf$_{pre-lesion}$ (purple) and Pf$_{pre-control}$ (gray). 2-way repeated measures ANOVA: Block 1, group: $F_{(1,18)} = 16$, *p = 0.0016, Block 2: group: $F_{(1,18)} = 6$, *p = 0.0071; Šídák's multiple comparison test Pf$_{pre-lesion}$ vs Pf$_{pre-control}$ late block 1,*p < 0.0001; late block 2, *p = 0.0028. **g** Same as E, but for VAL$_{pre-lesion}$ (green) and VAL$_{pre-control}$ (gray). 2-way repeated measures ANOVA: Block 1, group: $F_{(1,16)} = 1.9$, p > 0.05, Block 2: group: $F_{(1,16)} = 0.07$, p > 0.05. **h** Variability of initial direction for all trajectories. 2-way repeated measures ANOVA; Block 1, group:

$F_{(1,18)} = 12$, *p = 0.0078, Block 2: group: $F_{(1,18)} = 5$, *p = 0.0385; Šídák's multiple comparison test Pf$_{pre-lesion}$ vs Pf$_{pre-control}$ late block 1,*p = 0.0348. **i** Same as in (**h**) for VAL lesion groups. 2-way repeated measures ANOVA: Block 1, group: $F_{(1,16)} = 0.002$, p > 0.05, Block 2: group: $F_{(1,16)} = 0.84$, p > 0.05. **j** Peak speed of all trajectories. 2-way repeated measures ANOVA: Block 1, group: $F_{(1,18)} = 0.16$, p > 0.05, Block 2: group: $F_{(1,18)} = 2.2$, p > 0.05. **k** Same as in (**j**), for VAL lesion groups. 2-way repeated measures ANOVA: Block 1, group: $F_{(1,16)} = 4$, *p = 0.02, Block 2, group: $F_{(1,16)} = 5$, *p = 0.0392. **l** Overshoot of target for hit trajectories for Pf lesion groups. 2-way repeated measures ANOVA: Block 1, group: $F_{(1,16)} = 2.5$, p > 0.05. **m** Same as in (**l**), for VAL lesion groups. 2-way repeated measures ANOVA: Block 1, group: $F_{(1,16)} = 7$, *p = 0.0124. **f–m** Asterisks show ANOVA effect of group (lesion vs control) in block. Mean ± SEM with thick colored lines and shaded bounds. Pf$_{pre-lesion}$, n = 8 mice; Pf$_{pre-control}$, n = 10 mice; VAL$_{pre-lesion}$, n = 7 mice; VAL$_{pre-control}$, n = 10 mice. See also Supplementary Fig. 10. Source data are provided as a Source Data file.

not necessary for the execution of already learned reaches. To this end, we again ablated glutamatergic cells in Pf$_{FL}$ and VAL$_{FL}$ in Vglut2-Cre mice and used their WT littermates as controls, but only once they had learned to reach to the first target (Fig. 6a). Mice were implanted with a headcap and headpost and trained in block 1 of the STT for 28 consecutive days. After block 1 training, all mice were bilaterally injected with AAV5-flex-taCasp3-TEVp (AAV5) in either Pf$_{FL}$ or VAL$_{FL}$ (Pf$_{post-control}$ n = 5, Pf$_{post-lesion}$ n = 6, VAL$_{post-control}$, n = 6; VAL$_{post-lesion}$ n = 4) (Fig. S11a–c). After three-weeks mice were trained for an additional nine test days on the same target from block 1.

To examine how performance of directional reaches was affected by lesions after learning (Fig. 6b, c), we normalized all test day behavior to the last day of pre-lesion training and compared the animal's performance to this baseline. This analysis revealed that neither thalamic lesion affected the successful performance of directional reaches (hit ratio) (Fig. 6d, h), nor did they affect the engagement of the animals in the task directly after the lesion (number of attempts made) (Fig. S11d/e). Interestingly, Pf$_{post-lesion}$ mice significantly increased the variability of their initial reach direction compared to baseline at the end of the 9 day testing period (Fig. 6e), as well as increased the number of attempts made in a session (Fig. S11d). These effects were not seen in the control group or the VAL cohort (Fig. 6i and Fig. S11f–g). This may indicate that the lack of Pf neuronal activity gradually impaired the precision of the initial direction of reaches over many days of training. Still, we did not detect such a trend for the peak speed of movements or the target overshoot in the VAL animals (Fig. 6j, k) or the Pf animals (Fig. 6f, g). These results suggest that VAL is not needed post-learning to regulate reach speed once target reaches have been learned. However, post-learning Pf lesions led to long-term effects in the precision of learned directional reaches, suggesting that Pf activity may be needed to maintain low directional variability with continued training over long periods of time.

## Discussion

Through functional recording and manipulation experiments, we revealed that Pf$_{FL}$ and VAL$_{FL}$ have dissociable roles in the refinement of forelimb reaches to a spatial target. We mapped Pf$_{FL}$ and VAL$_{FL}$ as the thalamic areas that primarily target forelimb striatum and the CFA in M1, respectively. Pf$_{FL}$ and VAL$_{FL}$ neural populations were most active during reaching on early days, but the magnitude of their engagement decreased over learning, which was mediated through a change in the number and magnitude of reach-modulated cells. This result was further confirmed by tracking the same cellular populations over multiple days, where we found that matched cells in Pf decreased their activity during learning, and increased their modulation after a target switch. VAL matched cell populations were mostly stable throughout training, with subtle changes driven by negatively modulated cells. We also show, using PCA, that the dimensions capturing the dominant co-

activation patterns of the same matched neural populations in Pf$_{FL}$ shifted between early and late stages of learning, but stabilized late in learning. In contrast VAL$_{FL}$ dynamics shifted from day to day regardless of learning stage. Importantly, Pf$_{FL}$ neurons, more so than VAL$_{FL}$ cells, encoded directional information early in learning, and that directional information decreased over learning. In line with these findings, targeted pre-learning lesions of Pf$_{FL}$ and VAL$_{FL}$ revealed that Pf$_{FL}$, but not VAL$_{FL}$, is needed to refine directional variability of reaches and consequently learn the task. VAL$_{FL}$ lesions, on the other hand, did not affect the learning and refinement of the reaches, but affected the reach speed and led to animals overshooting the target. Overall, this data is aligned with previous findings showing that Pf and VAL have important roles in motor behavior[3,22,55,56], but reveals an exquisite dissociation in their roles in learning to refine directional reaches to a target.

General movement-related activity in Pf has been seen in mice during expert performance of learned behaviors such as lever pressing, rotarod, and reward-tracking assays[38,55,57]. Pf has also been shown to have an important role in goal-directed and motor learning paradigms[22,38,58]. Our work complements and expands on these findings by closely examining how Pf activity evolves over the learning process and what behaviorally relevant information can be decoded by neural populations.

One of our most striking results came from investigating if Pf or VAL neural activity encoded the direction of reaches. Other thalamic nuclei are already known to contain head direction cells, which encode animals' directional heading in the horizontal plane[59,60]. Other work has reported that Pf has a role in controlling the direction of head turning actions in a reward tracking task in mice[57]. Our work in the forelimb domain aligns with these findings, and expands on them to include Pf in regulating direction of reaching actions even while the animal is head-fixed, and that Pf contains reach direction tuned cells. Interestingly, our regression models showed that Pf activity encoded directional information best early in learning. The upstream source of this directional information may stem from input to Pf from the superior colliculus, which is thought to send an orienting signal[61,62]. Unpredicted or especially significant events cause short-latency responses in the superior colliculus and Pf, which can be quickly routed to the striatum by way of Pf[63]. These directional signals may then be reinforced in the striatum that could allow the refinement of reach direction with learning.

Consistent with this directional information in Pf$_{FL}$ neural populations, when we performed pre-learning lesions of Pf$_{FL}$ mice did not refine their reaching direction. These pre-learning lesions did not affect the variability of directional reaches early in training, indicating that Pf is not needed for the exploration or production of different reach directions. Rather, our findings support a function of Pf in conveying directional information to downstream structures that

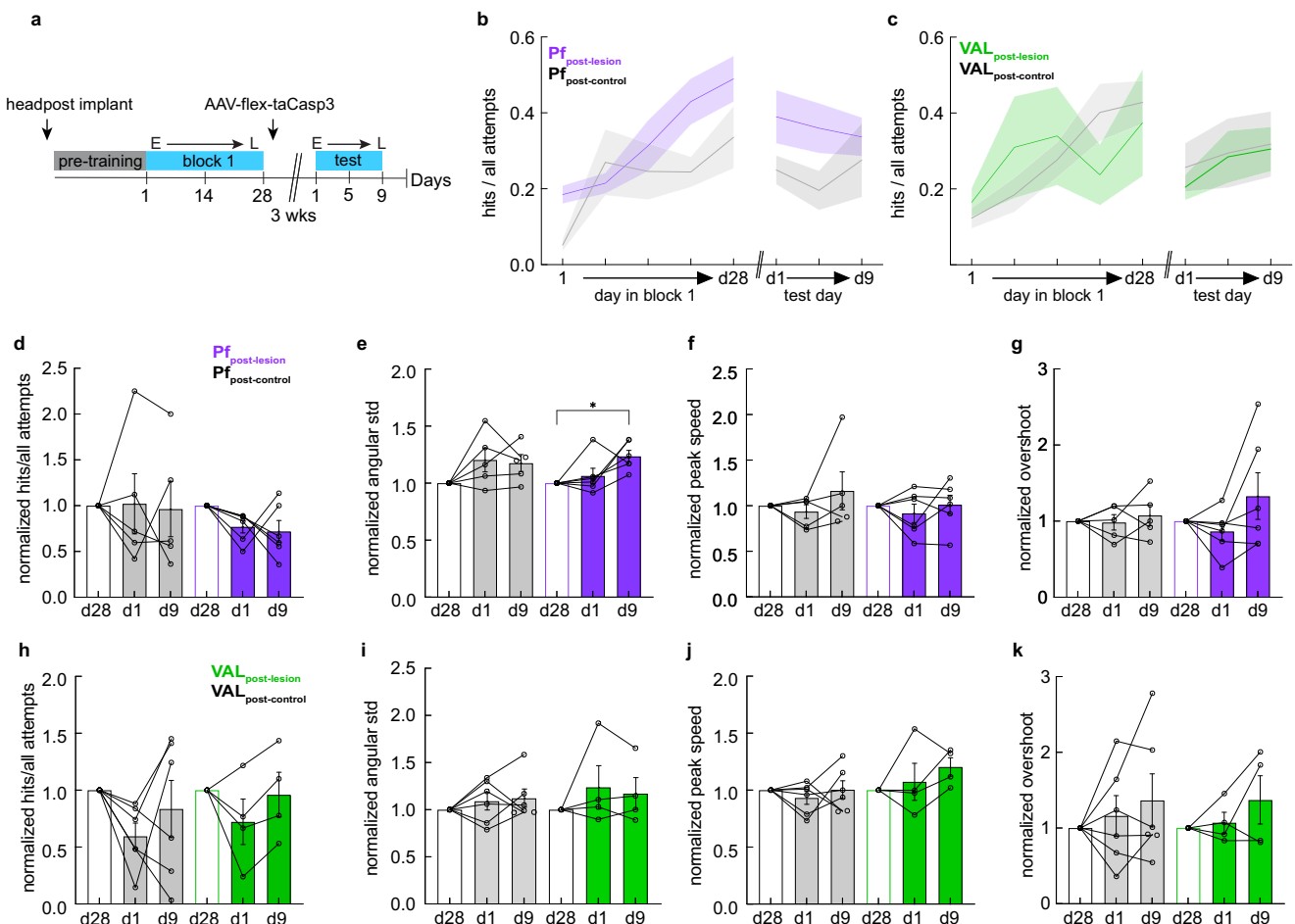

**Fig. 6 | Pf and VAL are not required to perform well-learned directional reaches.**
**a** Experimental design showing the timeline of surgeries, STT training, and testing. **b** Hit ratio on five selected days for block 1 and three test days post-lesion. Pf$_{post-lesion}$ (purple), n = 6 mice; Pf$_{post-control}$ (gray), n = 5 mice. Mean ± SEM in thick colored lines and shaded bounds. The same animals are used in (**d**–**g**). **c** Same as in (**b**), for VAL lesion groups. VAL$_{post-lesion}$ (green), n = 4 mice; VAL$_{post-control}$ (gray), n = 6 mice. The same animals are used in (**h**–**k**). **d** For Pf$_{post-lesion}$ (purple) and Pf$_{post-control}$ (gray) groups, normalized performance on the last day of block 1 before lesions (d28), and the first (d1) and last (d9) day of post-lesion test. Wilcoxon signed rank test, two sided, d9 vs d28: p > 0.05 for both groups. **e** For Pf$_{post-lesion}$ (purple) and Pf$_{post-control}$ (gray) groups, normalized variability of initial direction. Pf$_{post-lesion}$ mice increased their directional variability after lesions. Wilcoxon signed rank test, two sided, d9 vs d28: Pf$_{post-lesion}$, *p = 0.0312;

Pf$_{post-control}$, p > 0.05. **f** For Pf$_{post-lesion}$ (purple) and Pf$_{post-control}$ (gray) groups, normalized peak speed of movements. Wilcoxon signed rank test, two sided, d28 vs d9: p > 0.05 for both groups. **g** For Pf$_{post-lesion}$ (purple) and Pf$_{post-control}$ (gray) groups, normalized overshoot of target. Wilcoxon signed rank test, two sided, d28 vs d9: p > 0.05 for both groups. **h**–**k** Same as in (**d**–**g**) for VAL$_{post-lesion}$ (green) and VAL$_{post-control}$ (gray) groups. Wilcoxon signed rank test, two sided, d9 vs d28, p > 0.05 for all metrics and groups. **d**–**k** Performance metrics on the first (d1) and last (d9) day of post-lesion test. Data normalized to the values on the last day of training before lesion (d28, open white bars). Filled colored bars denote post-lesion days for Pf$_{post-lesion}$ (purple bars), VAL$_{post-lesion}$ (green bars), and control groups (gray bars). Asterisks indicate Wilcoxon test between d28 and d9, *p < 0.05. Mean ± SEM and single animals shown. See also Supplementary Fig. 11. Source data are provided as a Source Data file.

reinforce behavior, such as the striatum, early in learning, which would allow for successful direction commands to be reinforced through dopamine release upon reward delivery[64–66]. Additionally, as Pf pre-learning lesions did not affect learning to pre-training criterion, overall motivation, or engagement in the task, we do not think the effects on reaching were due to potential attentional deficits following lesions[67,68].

Later in training after refinement, when the reinforced direction has been learned and directional variability is low, Pf conveys less directional information. Interestingly, when we performed post-learning lesions of Pf$_{FL}$, mice were still able to perform with the same hit ratio, but after extended training post-lesion, the directional variability increased. This suggests that Pf activity is required for the maintenance and continuing reinforcement of the appropriate direction of forelimb reaches. These ideas align with recent findings that Pf inputs to the striatum are

needed to learn and maintain the performance of a sequential level pressing task[22], but differ in that we found Pf not needed for performance of our task once learned. This difference may be attributed to the differences in experimental design. In the study mentioned above, rats were trained to perform precisely timed lever presses and produced highly stereotyped behavior with time-varying kinematics coordinated across the full body. Our head-fixed reaching task relies on learning directional information and single-limb execution. It may be that Pf is more strongly involved in the performance of full-body behavior versus single-limb directional reaches. Alternatively, there may be different circuit requirements to execute a temporally precise action, which could require reinforcement of the Pf-DLS circuit on a faster timescale than directional reaching, and lead to a skill failing to persist without continuous Pf input. We did not observe any immediate changes in the timing of movements after Pf

lesions as the task itself does not rely on learned temporal execution, and does not lend itself to this type of analysis.

In VAL, we found that neural activity did not encode direction as strongly, nor was it needed to learn or perform targeted reaches. Other work has shown that VAL projections to M1 dynamically reorganize while training grasping movements[69], and are needed to perform cued grasping and lever-pulling tasks after learning[3,11,56]. With our focus on learning over time, we've added to these findings that VAL_{FL}'s low-dimensional activity changed regardless of learning stage, with individual cells changing their modulation from day-to-day. Persistent activity changes suggest that in our task, reorganization of VAL activity did not underlie learning. This was supported as pre-learning lesions of VAL did not produce a learning deficit. However, as our lesions left some areas of VAL intact, it is possible we would see different behavioral effects with larger VAL lesions. Previous work detailed that VAL input to M1 was needed for mice to learn a lever pulling task[56], which required head-fixed mice to pull and hold a lever for a set amount of time. The differences in our findings may be explained by task requirements. In our task, success was not contingent on the speed or timing of movements, and mice were not required to stop their movements inside of the target. Mice with pre-learning lesioning of VAL_{FL} were able to increase their hit ratio by learning the direction to the target, but moved faster and overshot the target more. As VAL integrates inputs from the DCN, these results align well with documented deficits in endpoint precision in human patients with cerebellar damage[20], as well as recent work in mice linking cerebellum to endpoint precision in reaching tasks[9,10,70]. There was also a subtle, but not statistically significant difference, in the hit ratio of VAL lesioned mice at the late stage of training, which may suggest that the final refinement needed for higher precision did not occur. Future experiments will explore whether VAL lesioned mice could learn a reaching movement that required more endpoint precision.

Our findings support an overarching view that basal ganglia related circuits are involved in motor learning through reinforcement learning, whereas the cerebellar related circuits are more important for movement initiation and error-based learning[71–73], and open the door for future studies to address the relative roles of subpopulations of Pf or VAL. As we did not target projection-specific subpopulations in our calcium imaging or ablation experiments, different subpopulations likely contributed to the observed results. Previous work showed movement-related activity following action initiation in striatum projecting Pf cells[55], suggesting that the reach-related modulation of cells we identified in Pf are striatum projecting as well. Additional findings demonstrated that striatum and STN projecting Pf cells have different roles in locomotion versus motor learning[38,74]. Follow-up investigations with projection-specific recordings and manipulations will be required to determine the roles of these subpopulations in the learning and performance of directional reaching, and may reveal even more detailed distinctions of Pf and VAL roles in reaching movements.

Taken together, our work shows that within the thalamus, there are multiple circuits governing separable components of motor behavior and refinement, namely direction and speed, with learning. Not only are Pf and VAL performing different functions, but these roles dynamically evolve with learning, which has important implications for understanding motor learning and movement-related disorders.

## Method
### Animals
All experiments and procedures were performed according to National Institutes of Health (NIH) guidelines, with lab protocols (AC-AABG2552, AC-AABG0559) approved by the Institutional Animal Care and Use Committee of Columbia University. Adult male mice (n = 10), aged 3–6 months, were used for all 2-photon calcium imaging experiments. Adult mice of both sexes (n = 26 female, n = 32 male), aged 2–6 months, were used for all lesion experiments. Adult mice of both sexes (n = 3), aged 4–8 months were used for anatomy experiments. The strains used were: C57BL6/J (Jackson Laboratories, strain #: 000664), VGlut2-Cre (Jackson Laboratories, strain #: 028863, C57BL6/J background). Prior to surgery, adult mice were housed in groups (2–5 per cage). Following surgery, mice used for behavioral experiments were individually housed with running wheels for enrichment. All mice were kept under a 12-h light–dark cycle.

### General surgical procedures
Analgesia was administered in the form of subcutaneous injection of Buprenorphine SR (0.5–1 mg/kg) on the day of surgery. Mice were anesthetized with isoflurane (1%–5%, plus oxygen at 1–1.5 l/min) and then placed in a stereotaxic holder (Kopf Instruments). The scalp was shaved and cleaned with 70% alcohol and iodine, and intradermal Bupivacaine (2 mg/kg) was administered. The mouse's body temperature was maintained throughout surgery at 37 °C using an animal temperature controller (ATC2000, World Precision Instruments). After surgery mice were allowed to recover until ambulatory in the home cage on a heating pad.

### Viral injections
For **anatomical tracing** experiments, a midline incision was made to expose the skull, and a craniotomy was made over the injection sites. Cholera toxin subunit B (Invitrogen lot #2384712, Alexa Fluor 555 conjugate; Invitrogen lot #2387465, Alexa Fluor 488 conjugate) reconstituted at 1% in HEPES buffered saline. Injections were made at the following volumes in respective areas (coordinates in mm from bregma): DLS at 0.75 AP, 2.45–55 ML, 2.45–5 DV: 80–100 nl; CFA at 1.5 AP, 2.3 ML, 0.55 DV: 45 nl.

**Imaging animals** were unilaterally injected with 200-300 nl AAV5-CamKII-GCaMP6f-WPRE.SV40 (UPENN − lot #CS1250, titer: 4.12 × 10^{12} GC/mL; or Addgene − lots #v23807 and v59618, titer: 2.3 × 10^{13} GC/mL), or AAV5-CAG-flex-GCaMP6f-WPRE.SV40 (Addgene − lot #v28551, titer: 3.5 × 10^{13} GC/mL) in Pf or VAL (Pf at −2.0 AP, 0.88 ML, 3.6 DV; VAL at −0.86 AP, 0.9 ML, 3.6 DV). A subset of animals were also injected with 60-100 nl retro-tdtomato virus pAAV-CAG-tdTomato (Retro) (ZIVirology custom virus− titer: 8.1 × 10^{12} vg/mL, lot #8) into either the forelimb-related area of DLS or the CFA. The expression of tdtomato was used for additional post-hoc verification of targeting forelimb projecting regions.

**Pre- and post-learning lesion** animals were bilaterally injected with 35–50 nl of AAV5-Ef1a-Flex-taCasp3-TEVp (UNC − lot #AV5760G, titer: 4.2 × 10^{12} vg/mL) in Pf or VAL at the same coordinates reported for imaging animals. All viruses and tracers were injected with the Nanoject III Injector (Drummond Scientific, USA).

### Headpost and tube rim implantation
All behavior animals followed the same pre and post-surgery protocol as described above. After mice were anesthetized and placed in a stereotaxic holder the scalp was removed to expose the skull. The fascia was cleared off and the cranium cleaned with saline and 3% hydrogen peroxide. A custom metal headpost (ZI Advanced Instrumentation platform) was secured to the skull using C&B Metabond dental cement (Parkell). The headbar was designed with small U-shaped ends on either side of the straight cemented bar that allowed easy sliding in and securing of the mouse's head in the head-fixation holders by tightening a screw through the U-shaped ends.

For post-learning lesion animals, mice underwent two surgeries. For surgery one, mice were implanted with a sterilized PCR tube rim (Axygen PCR strip tubes; REF: PCR-0212-FCP-C) and headpost. The tube rim was used to create a reference coordinate system to level the brain in surgery two, and to keep the skull above the injection target areas free from dental cement. The rim was placed centered around bregma, and secured with a combination of Metabond dental cement and superglue (Loctite). After the rim was secured, the headpost was

implanted. Measurements were taken between the rim's most anterior and posterior points aligned with bregma, two points of the headpost, and the marked anterior-posterior and medial-lateral coordinates for Pf or VAL. The exposed skull was again thoroughly sterilized, and then covered with a non-toxic sealant (Kwik-Sil silicone sealant). Surgery two took place after initial STT training. Because lambda was no longer visible after surgery one, the new coordinates measured at the end of the first surgery were used to level the mouse's skull in the stereotaxic frame. Craniotomies and viral injections were then made in the same manner as described above. Surgery two was completed by filling in the leveling rim with dental cement.

### Chronic GRIN lens implantation
After viral injection as described above, a 0.5 mm diameter gradient index (GRIN) lens (length: 6.51 mm, working distance: image side: 0 μm in air, object side: 500 μm in water, design wavelength: 920 nm, NA 0.5, non-coated, NEM-050-50-00-920-S-1.5p, GRINTECH) was implanted in the left thalamus (contralateral to the arm moving the joystick) above the injection site. Once in place, the lens was secured to the skull using a combination of Metabond dental cement and superglue. After GRIN lens implantation, headposts were implanted as described above.

### Behavioral setup
**SCARA joystick hardware and Spatial Target Task controls.** The SCARA (Selective Compliance Articulated Robot Arm) joystick and rig was built as described previously[43]. Animals were head-fixed in a custom-designed 3D printed cup (copyright IR CU21353), that we have previously shown allows increased workspace exploration compared to the standard tube. All animals used their right forelimb to interact with the SCARA joystick. A metal screw was placed horizontally to the left of the joystick to be used as arm rest for the left limb. The Spatial Target Task (STT) was controlled through a microcontroller board (Teensy 3.6, Arduino) and breakout board platform custom-built by the Advanced Instrumentation Platform at the Zuckerman Institute (TeenScience). The behavior task was written using the Arduino IDE. Briefly, the joystick was actively moved, or kept at the start position through a proportional integral derivative (PID) algorithm. The joystick position and all task events were recorded via serial output commands and recorded through Bonsai (OpenEphys).

### Spatial Target Task design and training
Animals were food restricted and given an individualized amount of chow food after each training session to maintain their body weight at 80% of pre-training baseline. Each session lasted until a maximum number of rewards (5–7 μl drop of 7.5% sucrose in water) were achieved, or the maximum time had elapsed.

During the session, the animal could move the joystick out of the start position (0/65 mm from the motor axis, 1 mm radius) and explore the workspace without any force generated by the motors for 7.5 s per attempt in a self-paced uncued manner.

The training schedule was designed as described previously[75] and included 2 pre-training phases, and 2 blocks of target training. Briefly, pre-training consisted of 4 days of Phase 1, and 5 or more days of Phase 2. In Phase 1 of pre-training, each initial touching of the joystick was rewarded (with a delay of 500 msec on days 1 and 2 and 1000 msec on days 3 and 4). Rewards were also given at random intervals between 5 and 15 s for continuous touching of the joystick, until 100 rewards were achieved.

For both Phase 2 and the target training, animals needed to move the joystick out of the start position and explore the workspace to receive a reward. If the mouse let go of the joystick for >200 msec, the attempt ended, the motors engaged and moved the joystick back to start (miss). If the criteria for a reward was met, the reward was delivered through the solenoid and the motors engaged and moved the joystick back to the start after a 750 msec delay (hit). In Phase 2 a

reward was given for moving the joystick in a forward direction of either a 40° or a 60° segment. Using the rewarded forward trajectories on the last day of Phase 2, 2 target locations were defined for each animal (Fig. S1b). The mean initial direction of all rewarded trajectories was calculated and a target center defined 40° to the left and right of the mean direction at 7.5 mm distance from the start position with a target radius of 3 mm for imaging animals, and 8 mm distance from the start position with target radius of 2.75 mm for lesion animals (Fig. S1a).

*Target training*: Target 1 was rewarded during the first block, and target 2 during the second block of target training. The reward was delivered immediately upon entering the target circle. A short ITI period, enforced by the SCARA motors, required the animal to stay in the start position between trials. The ITI could only be exited by exerting less than 11 g force against the joystick in any direction. After the ITI, each movement of the joystick out of the start position by the mouse was counted as an attempt. Task performance was calculated as the number of hits divided by all attempts (hit ratio). For imaging studies, when animals reached a 3-day average hit ratio > 0.6 or reached a maximum number of training days (53 days) on block 1, the target was changed in the next session and a new block began (performance criterion). For pre-learning lesion studies, maximum number of training days were 35 days in block 1 and 19 days in block 2. For post-learning lesion studies, mice were trained for 28 consecutive days before lesions and 9 additional days post-lesion on the same target.

### Behavioral training during 2-photon calcium imaging
To optimize head orientation under the 2-photon microscope, each mouse was trained with its own head-fixation setup, specifically adjusted for its headpost implantation. Pre-training took place in sound attenuating behavioral boxes for all mice. Mice were moved into the 2-photon behavioral rig after reaching over 0.5–0.65 hit ratio in pre-training. For both Pf and VAL cohorts, 2 mice each were moved into the 2-photon only after starting block 1. Performance often dropped once mice were moved into the 2-photon behavioral setup. Daily training sessions under the 2-photon microscope lasted until 50 or 120–150 rewards (on early and late days respectively) were achieved, or a maximum time had elapsed (60–80 min). For low performing animals, the target radius was increased early in block 1 or block 2, to up to 4.5 mm, but was reduced back to 3 mm by the end of the block.

### Quantification and statistical analysis
All data was analyzed using custom MATLAB code (MATLAB engine for python R2019b, Mathworks, Inc.) run from a Python analysis pipeline (Python 3.7.8) through a custom DataJoint[76] database (Datajoint 0.13.0). Statistical tests were performed with GraphPad Prism9 unless otherwise stated in methods. Data from the target training blocks are shown using 5 selected days per block. Each block includes an early and a late day, and 3 equidistant days in between. Two-way ANOVA with repeated measures with Šídák's correction, one-way ANOVA with repeated measures with Šídák's correction, paired and unpaired t-tests, Chi-squared test, and simple linear regression were used for statistical analysis. A *p* value of less than 0.05 was considered statistically significant.

### Analysis of spatial target task behavior
*Preprocessing of trajectories:* trajectories were down sampled to 6 msec intervals.

*Trajectories used for quantification:* For each attempt a joystick trajectory was recorded. All trajectories were used to quantify refinement of the movement. For hit trajectories, only the path from start to the point of target entry was used in the analysis of the average trajectory variance.

Behavior metrics were generated from raw joystick trajectories and analyzed as previously described[43]. Briefly, the *Occupancy* of the

workspace was quantified across $1 \times 1$ mm bins using the MATLAB 'histcount2' function on each trajectory. Dwell time in each bin and multiple visits to the same bin were discounted to calculate the total area visited, but are shown as heatmaps for an example animal (Fig. 1g). To calculate the *mean trajectory variability* trajectories were downsampled to 200 points. The mean trajectory was calculated by averaging the 200 coordinates of all trajectories. Then the standard deviation was calculated as the square root of the squared shortest distance to the mean trajectory at each point, divided by the number of trajectories. The average standard deviation along the 200 is reported.

The *initial vector variability* was calculated from initial vectors defined from the point at which the trajectory left the start circle (1 mm radius) to the point of it crossing a circle of 2.2 mm radius from the start position center. For each vector the angle was calculated using the 'atan2' MATLAB function. The angular standard deviation across all angles was calculated using the 'circ_std' function from the CircStat circular statistics toolbox[44], which is bounded between the interval $[0, \sqrt{2}]$. The *peak speed* was calculated on the downsampled (6 ms) trajectory, which was smoothed using a 30 msec moving average ('smooth' function in MATLAB). For analyzed hits, only the trajectory from the start until entering the target was considered. The maximum value of the smoothed speed of each trajectory was averaged across all trajectories. For neural predictions of peak speed through regression models, all full length trajectories were used, sampled at the imaging frequency and not smoothed.

To calculate the *target overshoot* the pathlength between the point of the trajectory entering the target and the end of the trajectory, when motors engaged (750 msec after entry), was measured. The average target overshoot pathlength per session is reported. For neural predictions of maximal distance through regression models, the maximally reached distance from the first point of leaving the start position for each full length trajectory was used.

## Behavioral variance matching
For all sessions, trials were subsampled such that the variance in each session matched the mean variance across all animal-sessions for the early and late days of block 1. For the initial vector angular standard deviation, the variance was matched to 0.8 (a.u.), for the peak speed, the variance was matched to 43.3 (mm/ms), and for the maximal distance, the variance was matched to 4.0 (mm), for all animals and sessions. Sessions with fewer than 50 trials were excluded from the regression analysis.

## Two-photon calcium imaging
All imaging experiments were conducted on a Bergamo II rotatable two-photon laser-scanning microscope (Thorlabs, Inc.). The system was conFig.d with 8 kHz resonant-galvo-galvo laser scanning mirrors and imaging frames of $512 \times 512$ pixels (corresponding to an area of 834 μm x 834 μm) were acquired at 30 fps. The system was equipped with two-channel fluorescence detection with amplified non-cooled GAsP photomultiplier tubes (PMTs). Emitted fluorescence was first directed to the PMTs, then split into "green" and "red" channels by a 565 nm sharp edge long-pass dichroic mirror. The green channel and red channel were subsequently filtered by a 525 nm/39 nm bandpass filter, and 593 nm /40 nm bandpass filter, respectively, before detection in the PMTs. The microscope was controlled via ThorImage 4.

Imaging was performed through a Nikon 16x, 0.8 NA water immersion objective placed over the implanted GRIN lens with the laser beam sized to fill the back aperture. A Coherent Chameleon Vision S tunable titanium-sapphire laser tuned to 920 nm with 75 fs pulses at 80 MHz repetition rate was used. Dispersion correction was adjusted to maximize fluorescent brightness as recorded under the objective. The imaging power was modulated through a Conoptics 350-105 Pockels Cell driven by a Conoptics 302 RM Amplifier for each imaging session to stay in the linear range of pixel intensity. The same

field of view was imaged across days of behavioral training by matching the depth of imaging across sessions. To identify imaging fields of view, head-fixed mice were positioned such that their GRIN lenses were aligned to the imaging objective. The depth at which GCaMP fluorescence signals were strongest was noted for each individual animal, and each following imaging session was done at that approximate identified depth.

Behavioral and imaging data were synchronized in two ways. Behavior events (attempt start, reward delivery, joystick touch) were recorded as TTL pulses together with the imaging frames through a National Instruments DAQ and saved to an h5 file using ThorSync (Thorlabs, Inc.). The h5 file datastream wasthen used to synchronize with the full behavioral data using custom MATLAB code. Further, a TTL pulse (-10 μs) from the microscope was sent directly to the TeenScience behavior acquisition board for each imaging frame, directly integrating the imaging frame timestamp into the behavior datastream.

## Calcium imaging data pre-processing
Calcium imaging data was motion corrected using non-rigid motion correction, cells were segmented and raw fluorescence traces extracted using Suite2p (0.10.0)[53]. Automatically classified cells were further manually curated using the Suite2p GUI and cells with a skew of less than 0.1 were excluded. Using custom MATLAB code, the neuropil signal was subtracted from the raw fluorescent traces using a coefficient of 0.7. The trace was then detrended by subtracting the baseline fluorescence. Baseline fluorescence was calculated on the full session's trace using a gaussian filter and minimum followed by maximum filtering. After detrending, individual neuron traces were z-scored across the entire imaging session. This z-scored trace was used for all following analyses.

## Population activity analysis
To compute peri-event trial-averaged activity, the neural activity of individual neurons was aligned to the event and then averaged across attempts. Three different event windows were defined: pre-movement (−500 to 0 ms from movement onset), during movement (0–500 ms from movement onset), and after hit (0–500 ms from target hit). If the trial-averaged trace for a cell was above or below 3 x SD of that cell's baseline trace (trial-average from -600 – -433 ms before movement onset) for over 90 ms, that cell was classified as significantly up- or down-modulated, respectively. Each cell was classified as significantly modulated or not during each of the three time windows (pre-movement, during movement, and after hit). To compare the average population activity during the movement across sessions of the block, the trial-averaged activity of all cells from all animals of the Pf or VAL groups were pooled and averaged.

## Tracking cells across multiple imaging sessions
Cell matching was performed across two pairs of sessions; a late day of block 1 to the early day of block 2, and across two late days in block 1. We imaged the same field of view across experimental sessions based on landmarks in the field of view, and developed a method to identify which cells in a chosen session were present in the other session. Cell matching was performed using the motion-corrected, pre-processed, and curated region of interest (ROI) masks from the Suite2p analysis. One session was chosen as the "template session", and the ROIs of the other session were aligned to this template session. Specifically, we used Suite2p's non-rigid motion correction method to align the average image of each session to the average image of the template session, and applied the corresponding alignment shifts to the ROIs identified in the session. Then for all sessions, we calculated the percent-overlap between each ROI in the template session and the most overlapping ROI in the other session. We used each ROI's binary mask, i.e., the image of the ROI where each pixel had the value 1 if it was

part of the ROI and the value 0 if not, and we calculated the percent-overlap between two ROIs (ROI A and ROI B) as the average of two quantities: (1) the percentage of ROI A's pixels that are also ROI B's pixels, and (2) the percentage of ROI B's pixels that are also ROI A's pixels. We determined that an ROI on one session was matched to an ROI on another session if the percent-overlap exceeded 55%. This 55% threshold was based on our visual inspection of example sessions across animals (Fig. S4). Thus, this procedure determined if each ROI from a template session matched to a ROI from another session. Finally, we applied this procedure to calculate which ROIs on the template session were present on the other chosen session and used only those cells for analysis. Cell matching resulted in the following numbers of matched cells: late block 1 vs. early block 2: Pf: n = 116 cells, 4 mice, VAL: n = 86 cells, 5 mice; late block 1 vs. late block 1: Pf: n = 108 cells, 4 mice, VAL: n = 72 cells, 4 mice.

Two early days: Pf: n = 113 cells, 5 mice, VAL: n = 46 cells, 4 mice. Two late days: Pf: n = 114 cells, 5 mice, VAL: n = 72 cells, 5 mice. Switch days (late block 1 > early block 2): Pf: n = 116 cells, 4 mice, VAL: n = 86 cells, 5 mice.

## Principal component analysis for matched cells

For visualization of neural activity (Fig. 3e, f and Fig. S7a-b), principal component analysis (PCA) was performed on matched cells from over days, combined over all animals of the group. PCA was used to calculate the dimensions in which neural population activity most varied within neural activity space. In pre-processing, each cell's fluorescence was de-trended (see Methods) and z-scored based on the fluorescence activity across the entire session. We extracted movement-locked neural activity as the trial-averaged activity, i.e., the peri-event time histogram or PETH, in the temporal window starting 2 s before movement onset and ending 2 s after movement onset, creating a $N_{frame}$ x $N_{cell}$ matrix per day. For the visualizations in Fig. 3d–h and Fig. S7a-b, the matrices from the days with matched cells were vertically concatenated to build the full matrix $X$. PCA was performed on matrix $X$ using the MATLAB function 'pca'. The returned scores of the first 3 PCs were plotted separately for the first half of the frames (corresponding to the first day for cell matching) and the second half of the frames (corresponding to the second day for cell matching).

We then assessed whether the dimensions of neural variance were more different across two conditions than expected within a condition, separately analyzing each animal's simultaneously recorded neural population. We subsampled trials so that each condition had the same number of trials (i.e., the number of trials in the condition with fewer trials), and we randomly divided each condition into two halves of trials. For each condition-half, we calculated the movement-locked PETH, the covariance matrix Σ between the $N$ neurons across time based on the PETH, and performed PCA on the covariance matrix to identify the dimensions that capture the most variance. We extracted the covariance within the 3 dimensions capturing the most variance, i.e., within the subspace spanned by the top 3 principal components (PCs). To do this, we calculated the singular value decomposition (SVD) of the covariance matrix $\Sigma = USU^T$ where $U \in R^{N \times N}$ contains the PCs as column vectors and $S \in R^{N \times N}$ is a diagonal matrix containing the singular values which are the variance within each PC. The covariance within the top 3 PC's was calculated as $\Sigma_3 = U_{1:3}S_3U_{1:3}^T \in R^{N \times N}$, where $U_{1:3} \in R^{N \times 3}$ is the matrix containing the top 3 PCs as column vectors and $S_3 \in R^{3 \times 3}$ is the diagonal matrix containing the singular values corresponding to the top 3 PC's. We then calculated an alignment index of this low-dimensional covariance between the two condition-halves within a condition and compared it to the alignment between condition-halves across conditions.

The alignment index was the fraction of low-dimensional variance in condition-half $A$ captured in the subspace of condition-half $B$. We projected the low-dimensional covariance of condition-half $A$ into the

subspace of condition-half $B$ as $P_B\Sigma_{3,A}P_B^T$, where $P_B = U_{1:3,B}U_{1:3,B}^T \in R^{N \times N}$ is the projection matrix into the subspace spanned by condition-half $B$'s top 3 PC's. We then calculated the fraction of variance captured in the subspace as: $trace(P_B\Sigma_{3,A}P_B^T)/trace(\Sigma_{3,A})$. Our alignment index took the average of the alignment of $A$ with $B$ and the alignment of $B$ with $A$, as these two calculations are not equal in general. (For intuition of this asymmetry, we consider an example with two dimensions. Imagine PC1 is the same for $A$ and $B$ but that PC2 is orthogonal for $A$ and $B$, and that the fraction of $A$ variance in PC1 is more than $B$. In this example, the fraction of low-dimensional variance $A$ in subspace $B$ is higher than the fraction of low-dimensional variance $B$ in subspace $A$, because $A$ concentrates more of its variance in the dimension that is shared across $A$ and $B$.) We calculated the alignment index between condition-halves within a condition (and then averaging over the two conditions) and across conditions (averaging over the two directions of alignment) for each animal, averaging over 100 folds of matching trials across conditions and dividing conditions into halves.

In addition, we calculated a shuffle alignment index (for both "within-condition" and "across-condition") to assess a chance rate of alignment. In order to preserve the temporal structure of population activity but randomize the subspace in which temporal activity evolved, we shuffled the neuron identity of the $B$ covariance matrix before calculating the PCA and performing the within-condition and across-condition alignments: $trace(P_{B-shuffle}\Sigma_{3,A}P_{B-shuffle}^T)/trace(\Sigma_{3,A})$. This shuffle alignment allowed us to assess whether alignments in the data were greater than expected by chance.

## Linear regression

Ridge regularized linear regression models were used to predict either the initial vector direction, the peak speed, or maximum distance on single trials from neural data, using a 166.66 ms (5 frames) window from movement onset, concatenated across cells (X = $N_{trials}(N_{roi}$ * 5 frames)). Additionally, the initial vector direction was also predicted from a 166.66 ms (5 frames) window of neural activity preceding the movement onset. Separate variance-matched behavior data was used for each behavior aspect. Data was split in 70% train and 30% held-out test trials. The train data was used to train the models (scikit-learn 'RidgeCV') and model selection was performed on the train data using 6-fold cross-validation to find the best hyperparameters. Model accuracy was determined on held-out test data. A total of 10 random train/test splits were used to train 10 separate models for each animal-session.

In addition to the models that predict a single behavioral aspect, multivariate models were trained to predict all 3 aspects, and all combinations of 2 aspects. Here all trials were variance matched for initial vector angular std.

## Calculation of coefficient of determination

The coefficient of determination (R²) was calculated from the predicted and true behavioral variable of all test trials across all models trained on different train/test splits. The coefficient of determination was calculated as (1 - u/v), where u is the regression sum of squares ((y_true - y_pred) ** 2).sum() and v is the residual sum of squares ((y_true - y_true.mean()) ** 2).sum(), as in the scikit-learn toolbox. For multivariate regression models, a separate coefficient of determination was calculated for each variable and the mean is reported.

## Direction tuning

For each session, the trials (hits and misses) were split into bins according to their initial direction vector (bin size = 60 degrees). The peak fluorescence (z-scored) was extracted from each cell's activity window (0–500 msec from reach onset) for each trial of each bin. One-way ANOVA ('anova1' MATLAB) was used to determine if the peak activity of a given cell was significantly different between trials with different initial vector directions. If the overall ANOVA was significant

(p < 0.05) a multiple comparison test ('multcompare' MATLAB) was performed. A cell was considered significantly tuned to a specific direction if the direction with the highest peak activity mean was significantly different than the peak activity mean of at least one other direction in the multiple comparison test. Using the matched ROIs across pairs of days, we also determined whether cells kept their direction tuning across days. All matched cells were classified according to 4 categories: Not direction tuned on either day (Not Tuned), tuned on one day but not the other (Lost Tuning), tuned on both days but for different directions (Unstable), tuned on both days for the same direction (Stable). Ratios across all animals for each category are reported.

### Immunohistochemistry

Mice were deeply anesthetized with isoflurane and transcardially perfused with 1x phosphate-buffered saline (PBS) followed by ice-cold 4% paraformaldehyde (PFA). Brains were extracted and post-fixed in 4% PFA overnight, and then transferred to PBS. Brains were sectioned into 50-75µm coronal sections using a vibratome (Leica vibratome VT1000). Sections were either stored free-floating in PBS at 4 °C, or serially mounted onto slides, dehydrated and stored at 4 °C until they were processed. All sections were first rinsed with PBS and permeabilized with 1–3% Triton in PBS. Immunostaining for GCamp on slide mounted sections was performed with primary antibodies (Anti-GFP Polyclonal Antibody, Alexa Fluor 488; ThermoFisher Scientific, catalog number # A-21311) diluted at 1:500 – 1:1000 for 1 day at 4 °C. Counterstains of DAPI (ThermoFisher Scientific, catalog number #62247) or NeuroTrace™ (ThermoFisher Scientific, catalog number #N21483) for lesion experiment tissues were performed after primary antibody staining, at a dilution of 1:1000 and 1:100, respectively. For free-floating sections, immunostaining was performed with the same dilution of primary antibodies, with incubations being 1 day or 3 h, respectively.

### Slide scanning and anatomical reconstructions

Serially mounted coronal sections were imaged with an AZ100 automated slide scanning microscope equipped with a 4 × 0.4-NA objective (Nikon). Image processing and analysis were done using BrainJ as previously described[49]. Briefly, sections were registered using 2D rigid-body registration. Ilastik was used to segment cell bodies and neuronal processes of each section. Representative images were imported and labeled to create a training set of cell bodies, neural processes, and background fluorescence for the algorithm across all images, which then created probabilistic assignment of detected features. The resulting probability masks and cell coordinates were aligned to the Allen Brain Reference Atlas (ABA) using Elastix. These aligned cell counts were then used to delineate the forelimb-specific subregions of Pf and VAL in the ABA. GRIN lens placements in imaging animals were manually assessed by matching the ABA, updated with the forelimb-related Pf and VAL, with coronal sections of stained tissue and marking the center lenses (Fig. S3a/b). Lesioned areas were manually delineated from NeuroTrace staining on all sections and registered and aligned to the updated ABA. Lesion volumes were visualized using BrainRender[77]. The percentage of ABA areas affected by the lesions were quantified using custom MATLAB code.

### Pre-learning cell ablation and behavioral training

Vglut2-Cre mice (n = 19) and WT littermates (n = 20) were used for all ablation experiments. Pf or VAL were bilaterally targeted AAV-flex-taCasp3-TEVp (AAV5), which triggers cell death through apoptosis[54]. All mice were injected with the flexed-taCasp3 virus, resulting in expression of caspase3 only in Vglut2-Cre mice. Stereotaxic surgeries with viral injections and headpost implantation took place as described above, with bilateral craniotomies and injections targeting Pf or VAL. Surgeries took place 3 weeks before STT training began. Mice were trained in the STT in block 1 until they reached high performance criteria (three consecutive day average of 65% hit ratio) or for a maximum 35 days, whichever came first. Mice were trained in block 2 until reaching high performance criteria, or for a maximum 19 days.

### Post-learning ablation and behavioral training

Post-learning lesion mice underwent surgeries as described above. Vglut2-Cre mice (n = 11) and WT littermates (n = 10) were used. The initial surgeries for headbar implantation took place one week before STT training began. All mice were trained in block 1 for 28 days, and had their second surgery of AAV-flex-taCasp3-TEVp (AAV5) injection the following day. Mice were given ad lib food the day before the surgery, and for three days post-surgery. After that, they returned to a food restriction regimen for three weeks, where they maintained 80–85% of their pre-surgery weight, to allow for the expression of the virus. Mice were then trained on the same target as block 1 for 9 consecutive test days.

## Materials availability

Further information and requests for reagents and mice should be directed to and will be fulfilled by the Lead Contact. This study did not generate new unique reagents or mice.

## Reporting summary

Further information on research design is available in the Nature Portfolio Reporting Summary linked to this article.

## Data availability

The data of this study are available from the authors upon request. Source data are provided with this paper.

## Code availability

All analysis were performed using publicly available software and custom scripts written in MATLAB R2019b (Mathworks) and Datajoint (Datajoint 0.13.0). The codes are available from the authors upon request.

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

## Acknowledgements

We would like to thank Luke Hammond and Humberto Avila, and Dr. Darcy Peterka from the Zuckerman Institute's Cellular Imaging Platform for guidance with analysis of the lesion volumes in Imaris and with BrainJ. We would also like to thank Dr. Darcy Peterka for his support and guidance when collecting 2-photon calcium imaging data. We would like to thank Ramin Khajeh for discussions regarding neural analysis. We would like to thank Mariana Correia and Drew Baughman for additional support through lab and mouse colony management. National Institutes of Health (NINDS) F31NS111853 (LJS). National Institutes of Health BRAIN Initiative (NINDS) K99NS126307 (ACM). National Institutes of Health (NINDS) K99NS128250 (VRA). National Institutes of Health (NIMH) F32MH118714 (VRA). National Institutes of Health (NINDS) R00NS114194 (JMM). National Institutes of Health BRAIN Initiative (NINDS) U19NS104649 (RMC). Swiss National Science Foundation Postdoc fellowship P2EZP3_172128 (ACM). Swiss National Science Foundation Postdoc fellowship P400PM_183904 (ACM). Simons-Emory International Consortium on Motor Control (RMC, ACM).

## Author contributions

L.J.S. designed the study, analyses, and wrote the manuscript, in collaboration with A.C.M. and R.M.C. L.J.S. and A.C.M. made figures and illustrations. L.J.S. performed surgeries. L.J.S. performed imaging experiments. L.J.S. and T.X.C. performed lesion experiments. A.C.M. and T.X.C. wrote task control code. L.J.S. and T.X.C. processed and analyzed brain tissue. L.J.S. and ACM analyzed mouse behavior data. L.J.S., A.C.M., and V.R.A. analyzed 2-photon calcium imaging data. J.M.M. advised on analysis of 2-photon calcium imaging data. All authors reviewed and edited the manuscript.

## Competing interests
