## [Transparent Peer Review file · Nature Communications]

Dissociable roles of distinct thalamic circuits in learning reaches to spatial targets

Corresponding Author: Dr Leslie Sibener

Version 0:

Reviewer comments:

Reviewer #1

(Remarks to the Author)

Summary:

This project explores the roles of Parafascicular (Pf) and Vento-Anterior/ Vento-Lateral (VAL) thalamus in the learning of a directional joystick moving task, employing in vivo calcium imaging of thalamic cell bodies, and nucleus-specific lesions. The authors find task-relevant activity in both nuclei, with Pf activity early in learning predictive of initial movement direction. Lesions of Pf pre-training impaired refinement of direction, while VAL lesions caused overshoots, and lesion of either after training did not meaningfully impact performance. Authors suggest these results show different roles for these thalamic nuclei in learning to reach.

The scope of the project seems appropriate, and several aspects of the project are noteworthy. The behavior seems robust and complex enough to engage the circuits under investigation, and task-related in vivo imaging of somatic Ca²⁺ activity in thalamic nuclei is technically challenging and uncommon. The dissociable effects of lesioning each area are very interesting, although persistence of expert performance after post-training lesions is not what might have been predicted based on previous lesion studies (see major points below), so perhaps the results deserve further discussion. Overall, this is a strong manuscript, with enthusiasm for the results only somewhat diminished by a lack of statistical clarity (see major points below).

Major points:

1. There is a lack of statistical clarity at several points in the paper and no reference to which statistical tests were performed to show significance referenced in the text or by asterisks in figures, only a list of statistical tests in the methods without describing which was used where. For an example on line 82, “the variability of their mean trajectory significantly decreased (Fig.1h).” But what is “significant” change in this context of figures 1f-k? While the interpretation of these data are quite intuitive, it would still be helpful to use some simple statistical tests to quantify the changes from early to late learning.
2. While the tasks, manipulations, and species are somewhat different, these results differ starkly from the results of the Olfeczký lab, which suggest thalamo-striatal input is still required for expert performance of a learned lever-pressing task in rats. Some discussion / explanation of this difference in findings seems appropriate.
3. A more complete anatomical description of Pf and VAL could be useful for strengthening the manuscript’s arguments, e.g. Line 42 and 43. “Distinct” how? Pf also receives input from cerebellar nuclei, so this isn’t a convincing argument for separate “anatomical footholds.” The “separateness” of the anatomy of VAL and Pf hasn’t been fully described here. The anatomy of their inputs and outputs are quite similar in several ways, despite the fact that they’ve been implicated in different brain functions and major differences not mentioned here: e.g. that their projections to cortex show very different anatomy and physiology. See also the minor point on Line 38 below.
4. Recorded cells are not projection-identified. It would be very interesting to know the projection identity (projecting to striatum or cortex or both) of the reach-related cells in Pf vs. non-reach-related. Thalamic sub-populations were mentioned briefly in discussion but are there any clues with regard to the Pf cells observed in this experiment? I do not necessarily think

projection-identification experiments are necessary but this is worthy of discussion. Ablation experiments are also not in projection-identified cells.

5. Pf neurons have also been shown to represent unexpected stimuli or stimuli requiring an unpredicted response (Minamimoto and Kimura, 2002) and contralateral visual neglect was reported to occur after lesions of the intralaminar nuclei including Pf in the cat (Orem et al. 1973), suggesting roles in attention and expectations for Pf. Is there a possibility that the deficits in learning due to Pf lesion are related to this attentional role? The results of the previous experiment showing Pf neural activity predictive of direction of movement is helpful in this regard, but it still seems likely that Pf lesion should cause other deficits not discussed here.

6. The changes in alignment index (Figure 3h-i) are an interesting idea, but I am finding them very difficult to follow/interpret. The idea that Pf activity stabilizes late in learning while VAL does not is an important idea, but I'm not sure how well these data support that conclusion. It seems strange to average the indices across animals. It would be nice to see these changes pairwise (dots connecting L/E or L/L values). Or perhaps the authors can either clarify their measure or show this effect in another way.

Minor points:

1. Line 38. "Primary inputs" - what is meant by primary? Pf is known to receive inhibitory inputs from cells of the BG, which may convey diverse information, but Pf also receives excitatory inputs from motor cortex and cerebellar nuclei, and these could also be argued to be "primary".

2. Line 60. "Critical nodes for skilled motor behavior" - these results that Pf and VAL are critical nodes for learning but not skilled behavior, right?

3. Figure 1f should say number of animals, shaded line is SEM etc.

4. Line 85: Fig 1j shows a mixture of animals increasing and decreasing peak speeds over training. If you mean average peak speeds across animals decreased over training, it should be shown that this is a statistically significant trend. It is also not clear what should be made of the individual differences.

5. 201: The basic properties of network dynamics in thalamus are presumably very different from in cortex where there is strong interconnection between the cells dependent on their distance from each other, and could be worth some discussion in its own right.

6. 278: Again, what statistical tests were performed here for figure 5f-m? Not listed in main text or figure legend. Also were these tests performed on the blank plots without asterisks? If so and the results were not significant, an "ns" or some other notation would be appropriate.

Line edits:

356: "nuclear" typo

361: "on early in learning" error

370: "their reaches direction" error

388: These lines are unclear. "This" in line 388 and 399, what is "this" referring to?

389: underlie (sp)

589: should be bold

606: sentence fragment

1058: "see also extended data figure"... omitted number of figure?

Reviewer #2

(Remarks to the Author)

In this manuscript, Sibener et al. describe the role of two nuclei of the thalamus (Pf and VAL) for learning a directional joystick reaching task in mice. Using 2-photon calcium imaging the authors find that Pf and VAL neurons are preferentially modulated during movement and their activity is progressively reduced over the course of learning. When switching the task to train mice to reach a second target only PF neurons undergo a similar adaptation from high to low activity, while VAL neurons show virtually no activity during this second block of training. Using dimension reduction analysis, the authors find that Pf population activity appears to adapt between early and late stages of learning but stabilizes with learning. VAL activity remains instable throughout learning. The study further finds that Pf encodes reach direction information during the

early training phase and lesioning Pf impairs the mouse's ability to refine their reach direction to learn the motor task. Overall, this study provides critical and novel findings on the role of thalamic nuclei for motor learning, which is an important and understudied question in motor neuroscience. However, the lack of some key analyses to support this study's claims limits my enthusiasm. While the authors have generated an impressive dataset of neural activity and behavioral data, the analysis of this data could be expanded on, and the current presented analysis appears to give the reader an incomplete picture of the neural adaptation in Pf and VAL during motor learning.

Major concerns

* The authors state that the Pf neuronal activity occupies different dimensions during early vs. late learning and is stabilized in late learning sessions. However, this argument is based on comparisons between late-stage training to one target and early-stage training to a second target. An alternative interpretation would be that the population activity is more similar when comparing different days of the same target as opposed to comparing two training days with different target locations. To claim that learning "stabilizes" the population activity dynamics an additional comparison of two early training days with the first target and a comparison of an early to a late training day with the same target would be necessary.

* It is a bit worrying that only ~35% of Pf and ~15% of VAL were lesioned for the loss of function experiments. On the one hand, it is maybe not surprising that such small lesions, especially in VAL, did not lead to strong behavior deficits; on the other hand, it is surprising that lesioning of even a very small portion of Pf had an effect on motor learning. It appears that even lesioning 5-10% of Pf significantly reduces the hit rate after learning (Fig. S6E). Are these lesions affecting a specific part of Pf that is particularly important for this task? A better characterization of the lesion in Pf and VAL might aid with interpreting these results.

* Statistical analysis for figure 3C and S5C is lacking. It appears that the fraction of newly negatively modulated cells has the most drastic difference between two late days vs. a late block 1 and an early block 2 day for both Pf and VAL neurons. Only mentioning the fraction of newly positively modulated cells for Pf and only newly negatively modulated cells for VAL seems arbitrary. Are these the only statistically significant differences between late/late and late/early?

* Does the activity amplitude of neurons that show the same type of modulation across late block 1 and early block 2 change between those two time points? Or is the increase in activity on early block 2 days in Pf primarily driven by newly positively modulated cells?

* The authors focus their quantification in Fig. 2I on movement activity, but how does overall activity at different phases of the task (pre-movement, movement, post-movement, reward) change with training. There appears to be quite a strong change in the activity at those pre and post movement phases.

I have a few additional minor concerns.

* Is the early block 2 training day used for the different analyses in Fig. 3 the "first" day of training with the new target? The text is not entirely clear. Also, Fig. 3 H and I are labeled as comparison "within day" and "across day"; are those comparisons indeed made within the same day or across days either within late block 1 or across late block 1 and early block 2? It should be across days. Please also double check the statistics presented for Figs. 3H and I (lines 228-239, are those the p-values for within vs across comparisons or L vs E and L vs L?).

* It would be helpful to show the behavior trajectories and quantifications on the chosen days for the cross-day analyses in Fig. 3 and S5

* For the reach direction encoding analysis the authors subsample "trials of all days to match the lowest reach direction variability". While I understand the need for this, I wonder what the implications are. What are the properties of these trials chosen by subsampling? It would be helpful to show their hit rate etc.

* Can Pf neural activity predict reach direction in early training with the second target?

* It would be helpful to show the outlines of Pf and VAL in the immunostaining images of Fig 5 B-E, so that the reader can gauge the lesion coverage.

Reviewer #3

(Remarks to the Author)

The thalamus is a central hub of the brain, interlinking distributed motor areas including the motor cortex, basal ganglia, and cerebellum. These motor areas likely have distinct roles during motor learning due to many factors, including the unique teaching signals they receive. Notably, different thalamic nuclei are connected with various parts of motor networks, making the thalamus an interesting brain area to monitor and manipulate activity to infer the functional roles of these networks.

Sibener et al. focused on the Pf and VAL nuclei, which receive input from the basal ganglia (BG) and deep cerebellar nuclei (DCN), respectively. Additionally, Pf projects back to the BG while VAL mainly projects back to the motor cortex (MCx). The study investigated how the activity of Pf and VAL changes during learning and tested their causality in learning through lesion experiments. For both recording and lesion experiments, significant differences between Pf and VAL were

discovered, which is intriguing and important for the field. However, there are several concerns that need to be addressed before publication, as listed below.

Major

1) Movement changes over training: As described in detail in Fig. 1, movement changes over the course of training, yet this has not been considered for major parts of the activity analysis (except in Fig. 4). Trial average activity could be high during learning because animals explore more speeds and locations. For example, these neurons might have a "receptive field" for particular locations or speeds of arm movement. If so, when mice explore various trajectories, in some trials, mice move within this receptive field, causing average activity to become higher. While trial averages are a good starting point, more sophisticated analyses considering differences in movement across trials are required to determine whether the activity in these thalamic nuclei changes independently of movement patterns during learning. Comparing activity in trials with similar trajectories and speeds across learning stages is one way to approach this. Alternatively, a regression model, like the one used in Fig. 4 (but not limited to initial movement direction), could be useful. A similar consideration of trial-by-trial movement is required for the PCA analysis in Fig. 3.

2) Fig. 4 analysis: If Pf activity "predicts" reach direction, it would be better to analyze the correlation between reach direction and activity before the onset of movement. Activity after the onset could be a mixture of sensory feedback, efference copy, etc, in addition to activity that drives movement.

3) Testing necessity during learning vs. expert in Fig. 6: It would be beneficial if the authors had retrained these mice for block 2 to confirm the lesions' effect on learning: then they can make a within animal comparison to conclude that these manipulations affect learning stronger than execution in expert mice. If the authors have performed such experiments, please report them.

4) Thalamic lesion effect on learned action: It would be beneficial if the authors had retrained these mice for block 2 to confirm the lesions' effect on learning. This approach would allow for a within-animal comparison to conclude that these manipulations affect learning more strongly than execution in expert mice. If the authors have performed such experiments, please report them.

Minor

5) Fig. 1f-k: These panels need statistics (apologies if I have missed them).

6) All figures: First letters should be capitalized.

7) Fig. 5b: It would be beneficial to have a high magnification view in control areas for comparison.

8) Fig. 5j-m: Controls do not seem to be consistent with Fig. 1. Any interpretation? This is why Fig. 1 requires statistics.

Version 1:

Reviewer comments:

Reviewer #1

(Remarks to the Author)

I am happy with the revisions. Thanks to the authors for their efforts on this nice paper.

Reviewer #2

(Remarks to the Author)

The authors have addressed all the concerns with additional analysis. Congrats on an important and elegant study.

Reviewer #3

(Remarks to the Author)

I appreciate that the authors have systematically addressed the reviewers' concerns, and the manuscript is much stronger. However, the results of the subsampling analysis to match trajectories do not appear to be mentioned in the manuscript (as far as I can tell). I believe it would be valuable to mention these results in 1-2 sentences in the main text (while agreeing with the authors' decision not to add additional figure panels).

We would like to thank the reviewers for their constructive and thoughtful comments on our submitted manuscript. In order to address the feedback received, we performed significant additional analysis to strengthen our main results and have improved the manuscript text for clarity and discussion. Specifically, we have updated Figures 1-5, and Extended Data Figures 6-11. Please find below our point-by-point responses to individual suggestions directly underneath the original comment. Our responses include descriptions and inclusion of new analyses, new figures, and updated text that address each comment for ease of reference, when appropriate. In the revised manuscript we have also included edits for clarity and conciseness; all changes from the original manuscript are highlighted in yellow.

Black text – editorial/reviewer original comments

Blue text – author responses

Green text – sections from revised manuscript

all changes from the original manuscript are highlighted in yellow.

REVIEWER COMMENTS

Reviewer #1 (Remarks to the Author):

Summary:

This project explores the roles of Parafascicular (Pf) and Ventro-Anterior/ Ventro-Lateral (VAL) thalamus in the learning of a directional joystick moving task, employing in vivo calcium imaging of thalamic cell bodies, and nucleus-specific lesions. The authors find task-relevant activity in both nuclei, with Pf activity early in learning predictive of initial movement direction. Lesions of Pf pre-training impaired refinement of direction, while VAL lesions caused overshoots, and lesion of either after training did not meaningfully impact performance. Authors suggest these results show different roles for these thalamic nuclei in learning to reach.

The scope of the project seems appropriate, and several aspects of the project are noteworthy. The behavior seems robust and complex enough to engage the circuits under investigation, and task-related in vivo imaging of somatic Ca²⁺ activity in thalamic nuclei is technically challenging and uncommon. The dissociable effects of lesioning each area are very interesting, although persistence of expert performance after post-training lesions is not what might have been predicted based on previous lesion studies (see major points below), so perhaps the results deserve further discussion. Overall, this is a strong manuscript, with enthusiasm for the results only somewhat diminished by a lack of statistical clarity (see major points below).

Major points:

1. There is a lack of statistical clarity at several points in the paper and no reference to which statistical tests were performed to show significance referenced in the text or by asterisks in figures, only a list of statistical tests in the methods without describing which was used where. For an example on line

82, “the variability of their mean trajectory significantly decreased (Fig.1h).” But what is “significant” change in this context of figures 1f-k? While the interpretation of these data are quite intuitive, it would still be helpful to use some simple statistical tests to quantify the changes from early to late learning.

We apologize for the missing values of statistical tests, which has led to the warranted concern in this comment. All statistical claims mentioned in the text were based on specific statistical tests, but were omitted from the legends. **We have corrected this mistake and added statistical information for all relevant panels in the legend of Figure 1** (on page 1 of Figure Document, and below), and all other figures.

Figure 1 : Mice explore and refine directional spatial target reaches

(a) Schematic of a head-fixed mouse performing the Spatial Target Task (STT) during imaging.

(b) Left: top view schematic of the Selective Compliance Articulated Robot Arm (SCARA) joystick showing the start position (green circle) and example position of target 1 (blue circle). Right: same as left, example position of target 2 (magenta circle) and previous target 1 (gray dashed area).

(c) Classification of trajectories from self-paced joystick movements. Hit trajectories (yellow line), and miss trajectories (gray line), started with the mouse moving the joystick out of the start position (green circle). Hit trajectories were rewarded upon target hit and ended by motors returning the joystick to the start 750 msec later. Miss trajectories were aborted by motors returning the joystick to the start if the animal let go of the joystick or the attempt timed out (after 7.5 sec).

(d) Schematic showing the phases of the training protocol: during 'pre-training' animals were rewarded first for touching the joystick, then for pushing it in a forward direction. During blocks 1 and 2, attempts that entered targets 1 or 2 were rewarded, respectively.

(e) Left: representative example trajectories on early and late days of block 1 with rewarded target (blue circle) and non-rewarding target (gray circle). Right: same for block 2 with rewarded target (magenta circle) and non-rewarding target (gray circle). Hit trajectories (yellow) that enter the rewarded target and miss trajectories (grey) are plotted in the two-dimensional workspace. Point of entering target and reward delivery is shown as small dark blue circle.

(f) Hit ratio on 5 selected days per block from early training day (E) to late training day (L) and 3 equidistant days in between for each animal. Mixed-effects model, day in block: $F(3,25) = 40$, $*p < 0.0001$, day x block: $F(3,14) = 4$, $*p < 0.05$. Šidák's multiple comparisons test: E vs L block 1, $*p < 0.005$; E vs L block 2, $*p < 0.0001$.

(g) Left: representative example heatmap of workspace occupancy of all trajectories of an early and late session (# times a bin is visited). Right: occupancy of workspace (mm^2). Mixed-effects model, day in block: $F(3,23) = 13$, $*p < 0.0001$, mouse: $F(1,9) = 6$, $*p < 0.05$. Šidák's multiple comparisons test: E vs L block 1, $*p < 0.05$; E vs L block 2, $*p < 0.005$.

(h) Variability of mean trajectory from all movements (averaged std along the length of the trajectory, mm). Mixed-effects model, day in block: $F(2,20) = 19$, $*p < 0.0001$, mouse: $F(1,9) = 17$, $*p < 0.01$. Šidák's multiple comparisons test: E vs L block 1, $*p < 0.005$; E vs L block 2, $*p < 0.005$.

(i) Variability of initial direction of movement (angular std, a.u). Mixed-effects model, day in block: $F(3,22) = 8$, $*p < 0.01$, mouse: $F(1,9) = 7$, $*p < 0.05$. Šidák's multiple comparisons test: E vs L block 1, $*p < 0.05$; E vs L block 2, $*p < 0.005$.

(j) Peak speed of trajectories (mm/sec). Mixed-effects model, day in block: $F(4, 36) = 4$, $*p < 0.05$, mouse: $F(1,9) = 8$, $*p < 0.05$. Šidák's multiple comparisons test: E vs L block 1, $p > 0.05$; E vs L block 2, $*p < 0.05$.

(k) Average target overshoot (mm) from target entry for all hit trajectories. Mixed-effects model, $F(3,25) = 8$, $*p < 0.001$, mouse: $F(1,9) = 9$, $*p < 0.05$. Šidák's multiple comparisons test: E vs L block 1, $p > 0.05$; E vs L block 2, $*p < 0.05$.

(f-k) Mean \pm SEM is shown in thick color lines and shaded bounds (block 1: blue, block 2: magenta), single animals are shown in gray lines ($n = 10$ mice).
See also Extended Data Fig. 1

2. While the tasks, manipulations, and species are somewhat different, these results differ starkly from the results of the Olveczky lab, which suggest thalamo-striatal input is still required for expert performance of a learned lever-pressing task in rats. Some discussion / explanation of this difference in findings seems appropriate.

The results from the Olveczky lab (Wolff et al., 2022) do differ from our results. We believe this is due to differences in the behavioral task requirements. Our task focuses on single-limb directional reaches that do not require precise timing. In Wolff et al., 2022, rats were trained to perform highly stereotyped movements that involved the coordination of precise timing across the entire body of a freely moving animal. **We have now elaborated on the differences and possible reasons for this observed divergence in the discussion.**

Page 10, lines 399-411

These ideas align with recent findings that Pf inputs to the striatum are needed to learn and maintain the performance of a sequential level pressing task²², but differ in that we found Pf not needed for performance of our task once learned. This difference may be attributed to the differences in experimental design. In the study mentioned above, rats were trained to perform precisely timed lever presses and produced highly stereotyped

behavior with time-varying kinematics coordinated across the full body. Our head-fixed reaching task relies on learning directional information and single-limb execution. It may be that Pf is more strongly involved in the performance of full-body behavior versus single-limb directional reaches. Alternatively, there may be different circuit requirements to execute a temporally precise action, which could require reinforcement of the Pf-DLS circuit on a faster timescale than directional reaching, and lead to a skill failing to persist without continuous Pf input. We did not observe any immediate changes in timing of movements in our task after Pf lesions, task itself does not rely on learned temporal execution, and does not lend itself to this type of analysis.

3. A more complete anatomical description of Pf and VAL could be useful for strengthening the manuscript's arguments, e.g. Line 42 and 43. "Distinct" how? Pf also receives input from cerebellar nuclei, so this isn't a convincing argument for separate "anatomical footholds." The "separateness" of the anatomy of VAL and Pf hasn't been fully described here. The anatomy of their inputs and outputs are quite similar in several ways, despite the fact that they've been implicated in different brain functions and major differences not mentioned here: e.g. that their projections to cortex show very different anatomy and physiology. See also the minor point on Line 38 below.

We agree that the anatomical descriptions of Pf and VAL can be better detailed so that their similarities and differences are more clearly described. **We have restructured an introductory paragraph on Pf and VAL anatomy to address this.**

Page 2, lines 35-45

"The thalamic parafascicular (Pf) and the ventroanterior/ventrolateral (VAL) nuclei in particular, are situated at the convergence of important motor centers. Pf receives inputs from basal ganglia output nuclei²³⁻²⁵, the superior colliculus^{26,27}, the cerebral cortex²¹, and sparse collaterals from deep cerebellar nuclei (DCN)²⁸. VAL receives inhibitory inputs from basal ganglia output nuclei as well^{25,29,30}, and receives dense excitatory afferents from DCN^{31,32} and motor cortex³³⁻³⁵. Pf and VAL's outputs differ significantly. Pf's glutamatergic outputs comprise the main sub-cortical input to the striatum, which are somatotopically organized²¹ and almost equal in amount to the motor cortex inputs to striatum^{36,37}. Pf outputs also target the subthalamic nucleus (STN)³⁸, another important movement center, and form reciprocal cortico-thalamic loops with limbic, associative, and somatosensory circuits^{21,27}. VAL's outputs, on the other hand, primarily target layer 1, layer 3, and layer 5 of premotor and motor cortex, but not the striatum^{29,39,40}"

4. Recorded cells are not projection-identified. It would be very interesting to know the projection identity (projecting to striatum or cortex or both) of the reach-related cells in Pf vs. non-reach-related. Thalamic sub-populations were mentioned briefly in discussion but are there any clues with regard to the Pf cells observed in this experiment? I do not necessarily think projection-identification experiments are necessary but this is worthy of discussion. Ablation experiments are also not in projection-identified cells.

We agree that there is need for more discussion on how projection identity may influence our results. While we did not record or manipulate projection-specific subpopulations, we targeted the region of Pf that projects to the forelimb-related DLS, as identified by retrograde tracings (Extended Data Fig. 2). A previous study has shown that DLS-projecting Pf cells were modulated after movement initiation (Díaz-Hernandez, et al., 2018). Similarly, in our Pf recordings, we found that about 80% of cells were movement-modulated as well, suggesting that our recordings were from equivalent populations. **We've added an additional paragraph on this topic in the discussion section of the paper.**

Page 11, lines 436-444

“As we did not target projection-specific subpopulations in our calcium imaging or ablation experiments, different subpopulations likely contributed to the observed results. Previous work showed movement-related activity following action initiation in striatum projecting Pf cells⁵⁶, suggesting that the reach-related modulation of cells we identified in Pf are striatum projecting as well. Additional findings demonstrated that striatum and STN projecting Pf cells have different roles in locomotion versus motor learning^{38,75}. Follow-up investigations with projection-specific recordings and manipulations will be required to determine the roles of these subpopulations in the learning and performance of directional reaching, and may reveal even more detailed distinctions of Pf and VAL roles in reaching movements.”

5. Pf neurons have also been shown to represent unexpected stimuli or stimuli requiring an unpredicted response (Minamimoto and Kimura, 2002) and contralateral visual neglect was reported to occur after lesions of the intralaminar nuclei including Pf in the cat (Orem et al. 1973), suggesting roles in attention and expectations for Pf. Is there a possibility that the deficits in learning due to Pf lesion are related to this attentional role? The results of the previous experiment showing Pf neural activity predictive of direction of movement is helpful in this regard, but it still seems likely that Pf lesion should cause other deficits not discussed here.

It is certainly possible that there are other deficits caused by our lesions to Pf we have not detected. However, we do not believe that the learning deficits we observe are due to Pf's role in motivation/attention. By analyzing the number of reach attempts made in the task (Extended Data Fig. 10c) we see that lesioned mice are not less active during the task, and still meaningfully engage in the task. Furthermore, as this task is self-initiated, performance does not rely on detecting “discriminatory” or “go” cues for success. Additionally, Pf lesion mice improve their performance during the pre-training phase of behavior (when mice need to push the joystick forward), which shows that they can learn simple action-outcome associations and are motivated by the reward.

Extended Data Fig. 10. Behavioral training of pre-learning Pf and VAL lesioned mice in spatial target task

(a) Hit ratio (hits / all attempts) of each animal in pre-learning lesion experiment. High performance criterion (65% hit ratio) indicated in dotted black line. $Pf_{pre-lesion}$ (purple), $n = 8$; $Pf_{pre-control}$ (black lines), $n = 10$.

(b) Same as (a) but for VAL pre-learning lesion experiment. $VAL_{pre-lesion}$ (green lines), $n = 7$; $Pf_{pre-control}$ (black lines), $n = 10$.

(c) Engagement (# of attempts) on the first day of block 1 and block 2. $Pf_{pre-lesion}$ vs $Pf_{pre-control}$, unpaired t -test; Block 1: $t(18) = 0.37$, $p > 0.05$, Block 2: $t(18) = 1$, $p > 0.05$. $VAL_{pre-lesion}$ vs $VAL_{pre-control}$, unpaired t -test, Block 1: $t(16) = 0.52$, $p > 0.05$, Block 2: $t(16) = 0.13$, $p > 0.05$.

As to the point of visual neglect, we do not think the behavior would be affected by this domain because the task does not use visual information as training is in the dark. To highlight the role of Pf in attention, **we've also added reference to attentional roles in our discussion**, with citations to Orem et al., 1973 and Minamimoto and Kimura, 2002.

Page 10, lines 385-393

“Consistent with this directional information in Pf_{FL} neural populations, when we performed pre-learning lesions of Pf_{FL} , mice did not refine their reaching direction. These pre-learning lesions did not affect the variability of directional reaches early in training, indicating that Pf is not needed for the exploration or production of different reach directions. Rather, our findings support a function of Pf in conveying directional information to downstream structures that reinforce behavior, such as the striatum, early in learning, which would allow for successful direction commands to be reinforced through dopamine release upon reward delivery⁶⁵⁻⁶⁷. Additionally, as Pf pre-learning lesions did not affect learning to pre-training criterion, overall motivation, or engagement in the task, we do not think the effects on reaching were due to potential attentional deficits following lesions^{68,69}.”

6. The changes in alignment index (h-i) are an interesting idea, but I am finding them very difficult to follow/interpret. The idea that Pf activity stabilizes late in learning while VAL does not is an important idea, but I'm not sure how well these data support that conclusion. It seems strange to average the indices across animals. It would be nice to see these changes pairwise (dots connecting L/E or L/L values). Or perhaps the authors can either clarify their measure or show this effect in another way.

We agree that our claims regarding changes in alignment index can be strengthened and clarified. To further investigate the stability of neural activity across pairs of days early and late in learning, we have added additional data **from two early days in block 1, two early days in block 2, and two late days in block 2 (Figure 3a-b)**. The data plotted in Figure 3g-h now reflect the alignment indices between two early or two late days (averaged over both blocks), and the change in alignment for the days around the block switch. **We've added pairwise comparisons in the figure, as well as paired t-tests for statistical evaluation.** These pair-wise comparisons show how aligned the neural activity is between days (closed circles), compared to a 'baseline' alignment within a single day (open circles). A significant difference in this comparison shows that neural activity changes across days more than within a single session, i.e. neural activity is unstable across days.

Our initial results are strengthened adding more matched days from both blocks. For Pf, the alignment index for matched cellular populations is different from day to day early in learning, and also during the block change (Figure 3g, data in blue and black), but stabilizes (is not different from 'baseline') in late performance (Figure 3g, data in magenta). For VAL, the alignment index changes daily, regardless of learning stage (Figure 3h).

Figure 3: Low dimensional population activity alignment changes between early and late learning

(a) Pf_{FL} trial-averaged fluorescence aligned to movement start for matched cells on two early days of block 1 (early 1 – black, early 2 – blue, $n = 113$ cells, 5 mice), two late days of block 1 (late 1 – pink, late 2 – cyan, $n = 114$ cells, 5 mice), two days around block switch (late 2 – cyan, early 1 – black, $n = 116$ cells, 5 mice), two early days of block 2 ($n = 67$ cells, 4 mice), and two late days of block 2 ($n = 49$ cells, 4 mice). Wilcoxon matched-pairs signed rank test results: early 1-2 (block 1) $*p < 0.0001$, late 1-2 (block 1) $p > 0.05$, late 2 (block 1) – early 1 (block 2) $*p < 0.0001$, early 1-2 (block 2) $*p < 0.0001$, late 1-2 (block 2) $p > 0.05$. Data shown as mean \pm SEM (thick colored line with shaded bounds).

(b) Same as in (a) for VAL_{FL}. Two early days of block 1 ($n = 46$ cells, 4 mice), two late days of block 1 ($n = 72$ cells, 5 mice), two days around block switch ($n = 86$, 5 mice), two early days of block 2 ($n = 97$ cells, 5 mice), and two late days of block 2 ($n = 103$ cells, 5 mice). Wilcoxon matched-pairs signed rank test results: early 1-2 (block 1) $*p < 0.001$. All other matched day pairs' Wilcoxon matched-pairs signed rank test: $p > 0.05$.

(c) Pf_{FL} population distribution of positively (white), negatively (gray), and non-modulated (dark gray) cells on all matched day pairs. Fisher's exact test (performed on cell counts): E1-E2 (block 1) $*p < 0.05$, E1-E2 (block 2) $*p < 0.05$. All other matched day pairs' Fisher's exact tests: $p > 0.05$.

(d) Same as in (c) for VAL_{FL}. Fisher's exact test (performed on cell counts) for all matched day pairs: $p > 0.05$.

(e) Neural activity of Pf_{FL} around movement start projected into top 3 PC space for late day of block 1 (cyan trace) and early day for block 2 (black trace). -2 seconds before movement onset (green circle), movement start (blue circle), and +2 seconds after movement start (red circle).

(f) Same as (d), for VAL_{FL} matched neuronal populations.

(g) For Pf_{FL} , average alignment of subspaces occupied within (open circle with dot) and across (filled circle) days with matched cells from two early days (E) in each block (black, average over block 1 and 2), two late days (L) in each block (magenta, average over block 1 and 2), and the block switch day from late block 1 to early block 2 (L/E, blue). For Pf_{FL} , paired t-test within vs across; E, $*p < 0.05$; L, $p > 0.05$; L/E, $*p < 0.005$.

(h) Same as (g), for VAL_{FL} matched cell populations. Paired t-test within vs across; E, $*p < 0.001$; L, $*p < 0.01$; L/E, $*p < 0.01$.

See also Extended Data Figs. 6-8

Minor points:

1. Line 38. “Primary inputs” - what is meant by primary? Pf is known to receive inhibitory inputs from cells of the BG, which may convey diverse information, but Pf also receives excitatory inputs from motor cortex and cerebellar nuclei, and these could also be argued to be “primary”.

We have rewritten this section of the introduction to more comprehensively describe the anatomical connections with Pf and VAL. We’ve removed the term “primary”.

Page 2, lines 33-45

“There is clear evidence that the thalamus, which is comprised of nuclei that receive inputs and send outputs to different motor centers, is a critical node in movement and motor learning^{21,22}. The thalamic parafascicular (Pf) and the ventroanterior/ventrolateral (VAL) nuclei in particular, are situated at the convergence of important motor centers. Pf receives inputs from basal ganglia output nuclei^{23–25}, the superior colliculus^{26,27}, the cerebral cortex²¹, and sparse collaterals from deep cerebellar nuclei (DCN)²⁸. VAL receives inhibitory inputs from basal ganglia output nuclei as well^{25,29,30}, and receives dense excitatory afferents from DCN^{31,32} and motor cortex^{33–35}. Pf and VAL’s outputs differ significantly. Pf’s glutamatergic outputs comprise the main sub-cortical input to the striatum, which are somatotopically organized²¹ and almost equal in amount to the motor cortex inputs to striatum^{36,37}. Pf outputs also target the subthalamic nucleus (STN)³⁸, another important movement center, and form reciprocal cortico-thalamic loops with limbic, associative, and somatosensory circuits^{21,27}. VAL’s outputs, on the other hand, primarily target layer 1, layer 3, and layer 5 of premotor and motor cortex, but not the striatum^{29,39,40}”

- Line 60. “Critical nodes for skilled motor behavior” - these results that Pf and VAL are critical nodes for learning but not skilled behavior, right?

In our original draft, we meant for the word “skilled” to refer to the reaching movement itself, not the training/performance stage (i.e. learning vs high performance). We have taken out the word “skilled” to avoid this confusion. Our results do show that Pf and VAL are important for learning, more so than continued performance after learning has occurred.

2. Figure 1f should say number of animals, shaded line is SEM etc.

As panels f-k in Figure 1 show data from the same 10 mice, we describe the n and plotting parameters at the end of the figure legend, highlighted below, to save space. We can add it to each panel if that is preferred.

Figure 1 : Mice explore and refine directional spatial target reaches

(a) Schematic of a head-fixed mouse performing the Spatial Target Task (STT) during imaging.

(b) Left: top view schematic of the Selective Compliance Articulated Robot Arm (SCARA) joystick showing the start position (green circle) and example position of target 1 (blue circle). Right: same as left, example position of target 2 (magenta circle) and previous target 1 (gray dashed area).

(c) Classification of trajectories from self-paced joystick movements. Hit trajectories (yellow line), and miss trajectories (gray line), started with the mouse moving the joystick out of the start position (green circle). Hit trajectories were rewarded upon target hit and ended by motors returning the joystick to the start 750 msec later. Miss trajectories were aborted by motors returning the joystick to the start if the animal let go of the joystick or the attempt timed out (after 7.5 sec).

(d) Schematic showing the phases of the training protocol: during 'pre-training' animals were rewarded first for touching the joystick, then for pushing it in a forward direction. During blocks 1 and 2, attempts that entered targets 1 or 2 were rewarded, respectively.

(e) Left: representative example trajectories on early and late days of block 1 with rewarded target (blue circle) and non-rewarding target (gray circle). Right: same for block 2 with rewarded target (magenta circle) and non-rewarding target (gray circle). Hit trajectories (yellow) that enter the rewarded target and miss trajectories (grey) are plotted in the two-dimensional workspace. Point of entering target and reward delivery is shown as small dark blue circle.

(f) Hit ratio on 5 selected days per block from early training day (E) to late training day (L) and 3 equidistant days in between for each animal. Mixed-effects model, day in block: $F(3,25) = 40$, $*p < 0.0001$, day x block: $F(3,14) = 4$, $*p < 0.05$. Šidák's multiple comparisons test: E vs L block 1, $*p < 0.005$; E vs L block 2, $*p < 0.0001$.

(g) Left: representative example heatmap of workspace occupancy of all trajectories of an early and late session (# times a bin is visited). Right: occupancy of workspace (mm^2). Mixed-effects model, day in block: $F(3,23) = 13$, $*p < 0.0001$, mouse: $F(1,9) = 6$, $*p < 0.05$. Šidák's multiple comparisons test: E vs L block 1, $*p < 0.05$; E vs L block 2, $*p < 0.005$.

(h) Variability of mean trajectory from all movements (averaged std along the length of the trajectory, mm). Mixed-effects model, day in block: $F(2,20) = 19$, $*p < 0.0001$, mouse: $F(1,9) = 17$, $*p < 0.01$. Šidák's multiple comparisons test: E vs L block 1, $*p < 0.005$; E vs L block 2, $*p < 0.005$.

(i) Variability of initial direction of movement (angular std, a.u). Mixed-effects model, day in block: $F(3,22) = 8$, $*p < 0.01$, mouse: $F(1,9) = 7$, $*p < 0.05$. Šidák's multiple comparisons test: E vs L block 1, $*p < 0.05$; E vs L block 2, $*p < 0.005$.

(j) Peak speed of trajectories (mm/sec). Mixed-effects model, day in block: $F(4, 36) = 4$, $*p < 0.05$, mouse: $F(1,9) = 8$, $*p < 0.05$. Šidák's multiple comparisons test: E vs L block 1, $p > 0.05$; E vs L block 2, $*p < 0.05$.

(k) Average target overshoot (mm) from target entry for all hit trajectories. Mixed-effects model, $F(3,25) = 8$, $*p < 0.001$, mouse: $F(1,9) = 9$, $*p < 0.05$. Šidák's multiple comparisons test: E vs L block 1, $p > 0.05$; E vs L block 2, $*p < 0.05$.

(f-k) Mean \pm SEM is shown in thick color lines and shaded bounds (block 1: blue, block 2: magenta), single animals are shown in gray lines ($n = 10$ mice).

See also Extended Data Fig. 1

- 3. Line 85: Fig 1j shows a mixture of animals increasing and decreasing peak speeds over training. If you mean average peak speeds across animals decreased over training, it should be shown that this is a statistically significant trend. It is also not clear what should be made of the individual differences.**

We apologize for the missing values of statistical tests. All statistical claims that were mentioned in the text were based on specific statistical tests, but were omitted from the legend. Our analysis on speed, where we measure the average peak speed across all trials, does have higher day to day and inter-animal variability than other metrics. However, when performing a mixed-effects model we see that the value is different across days in both blocks, and that it decreases over block 2. **We have added the statistics for this panel, as well as all other relevant panels in Figure 1f-k.** (See below)

I made a text edit to modify line 89:

"In addition to learning to reach in the correct direction, we find an overall decrease in the peak speed of reaches across animals (Fig. 1j), suggesting that they slowed down to accurately hit the target"

Figure 1 : Mice explore and refine directional spatial target reaches

(a) Schematic of a head-fixed mouse performing the Spatial Target Task (STT) during imaging.

(b) Left: top view schematic of the Selective Compliance Articulated Robot Arm (SCARA) joystick showing the start position (green circle) and example position of target 1 (blue circle). Right: same as left, example position of target 2 (magenta circle) and previous target 1 (gray dashed area).

(c) Classification of trajectories from self-paced joystick movements. Hit trajectories (yellow line), and miss trajectories (gray line), started with the mouse moving the joystick out of the start position (green circle). Hit trajectories were rewarded upon target hit and ended by motors returning the joystick to the start 750 msec later. Miss trajectories were aborted by motors returning the joystick to the start if the animal let go of the joystick or the attempt timed out (after 7.5 sec).

(d) Schematic showing the phases of the training protocol: during 'pre-training' animals were rewarded first for touching the joystick, then for pushing it in a forward direction. During blocks 1 and 2, attempts that entered targets 1 or 2 were rewarded, respectively.

(e) Left: representative example trajectories on early and late days of block 1 with rewarded target (blue circle) and non-rewarding target (gray circle). Right: same for block 2 with rewarded target (magenta circle) and non-rewarding target (gray circle). Hit trajectories (yellow) that enter the rewarded target and miss trajectories (grey) are plotted in the two-dimensional workspace. Point of entering target and reward delivery is shown as small dark blue circle.

(f) Hit ratio on 5 selected days per block from early training day (E) to late training day (L) and 3 equidistant days in between for each animal. **Mixed-effects model, day in block: $F(3,25) = 40$, $*p < 0.0001$, day x block: $F(3,14) = 4$, $*p < 0.05$. Šidák's multiple comparisons test: E vs L block 1, $*p < 0.005$; E vs L block 2, $*p < 0.0001$.**

(g) Left: representative example heatmap of workspace occupancy of all trajectories of an early and late session (# times a bin is visited). Right: occupancy of workspace (mm²). **Mixed-effects model, day in**

block: $F(3,23) = 13$, $*p < 0.0001$, mouse: $F(1,9)=6$, $*p < 0.05$. Šídák's multiple comparisons test: E vs L block 1, $*p < 0.05$; E vs L block 2, $*p < 0.005$.

(h) Variability of mean trajectory from all movements (averaged std along the length of the trajectory, mm). Mixed-effects model, day in block: $F(2,20) = 19$, $*p < 0.0001$, mouse: $F(1,9) = 17$, $*p < 0.01$. Šídák's multiple comparisons test: E vs L block 1, $*p < 0.005$; E vs L block 2, $*p < 0.005$.

(i) Variability of initial direction of movement (angular std, a.u). Mixed-effects model, day in block: $F(3,22) = 8$, $*p < 0.01$, mouse: $F(1,9) = 7$, $*p < 0.5$. Šídák's multiple comparisons test: E vs L block 1, $*p < 0.05$; E vs L block 2, $*p < 0.005$.

(j) Peak speed of trajectories (mm/sec). Mixed-effects model, day in block: $F(4, 36) = 4$, $*p < 0.05$, mouse: $F(1,9) = 8$, $*p < 0.05$. Šídák's multiple comparisons test: E vs L block 1, $p > 0.05$; E vs L block 2, $*p < 0.05$.

(k) Average target overshoot (mm) from target entry for all hit trajectories. Mixed-effects model, $F(3,25) = 8$, $*p < 0.001$, mouse: $F(1,9) = 9$, $*p < 0.05$. Šídák's multiple comparisons test: E vs L block 1, $p > 0.05$; E vs L block 2, $*p < 0.05$.

(f-k) Mean \pm SEM is shown in thick color lines and shaded bounds (block 1: blue, block 2: magenta), single animals are shown in gray lines ($n = 10$ mice).

See also Extended Data Fig. 1

- 4. 201: The basic properties of network dynamics in thalamus are presumably very different from in cortex where there is strong interconnection between the cells dependent on their distance from each other, and could be worth some discussion in its own right.**

It is true that the microcircuitry of the thalamus is quite different from that of cortical networks. Mainly, while the cortex has a large amount of local recurrent connectivity, the thalamus does not have local connectivity within individual nuclei. Because VAL's main output is to the motor cortex, it would be interesting to examine the relationship between VAL and M1's dynamic differences. An interesting study (Sauerbrei et al., 2019, Nature) showed that M1's dynamics during a forelimb task in mice were driven by input from the motor thalamus. We cite this paper in our introduction, but think that an additional discussion of differences between cortex and thalamus are beyond the scope of this manuscript. Since we did not record or manipulate the cortex we would rather not want to distract too much from our main discussion around Pf and VAL, even though this is a very interesting line of thought.

- 5. 278: Again, what statistical tests were performed here for figure 5f-m? Not listed in main text or figure legend. Also were these tests performed on the blank plots without asterisks? If so and the results were not significant, an "ns" or some other notation would be appropriate.**

We apologize again for the missing statistical values in the legend. 2-way repeated measures ANOVA analyses were performed for all panels in Figure 5, including the ones without asterisks. We have now noted all non-significant comparisons in the legend as well, for p values over 0.05.

Figure 5. Ablation of Pf_{FL} but not VAL_{FL} cells before learning impairs the refinement of initial reach direction

(a) Experimental design showing the timeline for surgeries and STT training, and coronal ABA sections showing the injection targets for Pf_{FL} and VAL_{FL} bilateral lesions.

(b) Representative coronal section of VGLut2-Cre mouse injected with AAV expressing taCasp3 - NeuroTrace (yellow) targeted to Pf_{FL} . Dashed-line indicating Pf. High-magnification view from the inset in box.

(c) Representative coronal section of WT-littermate mouse injected with AAV expressing taCasp3 - NeuroTrace (yellow) targeted to Pf_{FL} . Dashed-line indicating Pf. High-magnification view from the inset in box.

(d-e) Same as (b-c), for AAV injection targeting VAL_{FL} . Dashed-lines indicating VAL.

(f) Hit ratio (hits / all attempts) on five selected days over two blocks; Early (e) day of block, Late (L) day of block, and three equidistant days in between. $Pf_{pre-lesion}$ (purple) and $Pf_{pre-control}$ (gray). 2-way repeated measures ANOVA: Block 1, group: $F(1,18) = 16$, $*p < 0.001$, Block 2: group: $F(1,18) = 6$, $*p < 0.05$; Šidák's multiple comparison test $Pf_{pre-lesion}$ vs $Pf_{pre-control}$ late block 1, $*p < 0.0001$; late block 2, $*p < 0.01$.

(g) Same as E, but for $VAL_{pre-lesion}$ (green) and $VAL_{pre-control}$ (gray). 2-way repeated measures ANOVA: Block 1, group: $F(1,16) = 1.9$, $p > 0.05$, Block 2: group: $F(1,16) = 0.07$, $p > 0.05$.

(h) Variability of initial direction (angular std) for all trajectories. 2-way repeated measures ANOVA; Block 1, group: $F(1,18) = 12$, $*p < 0.01$, Block 2: group: $F(1,18) = 5$, $*p < 0.05$; Šidák's multiple comparison test $Pf_{pre-lesion}$ vs $Pf_{pre-control}$ late block 1, $*p < 0.01$; late block 2, $*p < 0.05$.

(i) Same as in (h) for VAL lesion groups. 2-way repeated measures ANOVA: Block 1, group: $F(1,16) = 0.002$, $p > 0.05$, Block 2: group: $F(1,16) = 0.84$, $p > 0.05$.

(j) Peak speed of all trajectories (mm/sec). 2-way repeated measures ANOVA: Block 1, group: $F(1,18) = 0.16$, $p > 0.05$, Block 2: group: $F(1,18) = 2.2$, $p > 0.05$.

(k) Same as in (j), for VAL lesion groups. 2-way repeated measures ANOVA: Block 1, group: $F(1,16) = 4$, $*p = 0.05$, Block 2, group: $F(1,16) = 5$, $*p < 0.05$.

(l) Overshoot (mm) of target for hit trajectories for Pf lesion groups.

(m) Same as in (l), for VAL lesion groups. 2-way repeated measures ANOVA: Block 1, group: $F(1,16) = 7$, $*p < 0.05$; Šidák's multiple comparison test $VAL_{pre-lesion}$ vs $VAL_{pre-control}$ late block 1, $*p < 0.05$.

(f-m) 2-way ANOVA with repeated measures, asterisks show ANOVA effect of group (lesion vs control) in block. Panels without asterisks have no significant group effect. Mean \pm SEM with thick colored lines and shaded bounds.

See also Extended Data Fig. 9

Line edits:

356: “nuclear” typo

We have resolved this typo. The sentence now reads:

Page 10, line 374

“Other thalamic **nuclei** are already known to contain head direction cells, which encode animals' directional heading in the horizontal plane^{60,61}”

361: “on early in learning” error

We have resolved this typo and deleted the word “on” and updated this sentence for new results. The sentence now reads:

Page 9, line 359-360

“Importantly, Pf_{FL} **neurons, more so than VAL_{FL} cells**, encoded directional information early in learning, and that directional information **decreased over** learning.”

370: “their reaches direction” error

We have resolved this typo and corrected “reaches” to “reaching”. The sentence now reads:

Page 10, lines 385-386

“Consistent with this directional information in Pf_{FL} neural populations, when we performed pre-learning lesions of Pf_{FL} , mice did not refine **their reaching** direction.”

388: These lines are unclear. “This” in line 388 and 399, what is “this” referring to?

We have clarified the language in these two sentences. The lines now read:

Page 10-11, lines 417-418

“Persistent activity changes suggest that in our task, reorganization of VAL activity did not underlie learning.”

389: underlie (sp)

We have fixed this typo. The line now reads:

Page 10-11, lines 417-418

“Persistent activity changes suggest that in our task, reorganization of VAL activity did not underlie learning.”

589: should be bold

We have bolded this subheading within the methods section.

Page 16, line 620

Behavioral training during 2-photon calcium imaging

606: sentence fragment

This line now reads as below:

Page 16, line 637

“Each block includes an early and a late day, and 3 equidistant days in between.”

1058: “see also extended data figure”... omitted number of figure?

All references to extended data figures have been corrected and updated in the text.

Reviewer #2 (Remarks to the Author):

In this manuscript, Sibener et al. describe the role of two nuclei of the thalamus (Pf and VAL) for learning a directional joystick reaching task in mice. Using 2-photon calcium imaging the authors find that Pf and VAL neurons are preferentially modulated during movement and their activity is progressively reduced over the course of learning. When switching the task to train mice to reach a second target only PF neurons undergo a similar adaptation from high to low activity, while VAL neurons show virtually no activity during this second block of training. Using dimension reduction analysis, the authors find that Pf population activity appears to adapt between early and late stages of learning but stabilizes with learning. VAL activity remains instable throughout learning. The study further finds that Pf encodes reach direction information during the early training phase and lesioning Pf impairs the mouse's ability to refine their reach direction to learn the motor task. Overall, this study provides critical and novel findings on the role of thalamic nuclei for motor learning, which is an important and understudied question in motor neuroscience. However, the lack of some key analyses to support this study's

claims limits my enthusiasm. While the authors have generated an impressive dataset of neural activity and behavioral data, the analysis of this data could be expanded on, and the current presented analysis appears to give the reader an incomplete picture of the neural adaptation in Pf and VAL during motor learning.

Major concerns

- 1. The authors state that the Pf neuronal activity occupies different dimensions during early vs. late learning and is stabilized in late learning sessions. However, this argument is based on comparisons between late-stage training to one target and early-stage training to a second target. An alternative interpretation would be that the population activity is more similar when comparing different days of the same target as opposed to comparing two training days with different target locations. To claim that learning "stabilizes" the population activity dynamics an additional comparison of two early training days with the first target and a comparison of an early to a late training day with the same target would be necessary.**

We agree that there was a need for extra day comparisons to fully support our claims and thank the reviewer for the suggestion.

We have added cell tracking across three more pairs of days so that we now show five pairs of matched days (**Fig. 3a-b**); two early days in block 1 and 2, two late days in block 1 and 2, and two days spanning block switch. We show trial averaged activity traces from all matched cells in Fig. 3a-b, and have added additional modulation classification for these tracked cells in Fig. 3c. **For the alignment index analysis**, the data plotted in Figure 3g-h now reflect the alignment index across the early and late days (averaged over both blocks for simplicity), and the change in alignment for the days around the block switch.

We've added lines connecting animal-sessions to indicate pairwise comparisons in the figure, and are using paired t-tests for statistical evaluation.

We were not able to perform the analysis with an early to late training day of the same target, as the reviewer suggested, because we did not get enough matched cells between recording sessions separated as much in time.

However, our findings in the original manuscript are strengthened by adding more matched days. For Pf, the alignment index for matched cellular populations is different from day to day early in learning and when the target is switched (Figure 3g, data in blue and black), but stabilizes in late performance (Figure 3g, data in magenta). For VAL, the alignment index changes day to day, regardless of learning state (Figure 3h).

Figure 3: Low dimensional population activity alignment changes between early and late learning

(a) Pf_{FL} trial-averaged fluorescence aligned to movement start for matched cells on two early days of block 1 (early 1 – black, early 2 – blue, $n = 113$ cells, 5 mice), two late days of block 1 (late 1 – pink, late 2 – cyan, $n = 114$ cells, 5 mice), two days around block switch (late 2 – cyan, early 1 – black, $n = 116$ cells, 5 mice), two early days of block 2 ($n = 67$ cells, 4 mice), and two late days of block 2 ($n = 49$ cells, 4 mice). Wilcoxon matched-pairs signed rank test results: early 1-2 (block 1) $*p < 0.0001$, late 1-2 (block 1) $p > 0.05$, late 2 (block 1) – early 1 (block 2) $*p < 0.0001$, early 1-2 (block 2) $*p < 0.0001$, late 1-2 (block 2) $p > 0.05$. Data shown as mean \pm SEM (thick colored line with shaded bounds).

(b) Same as in (a) for VAL_{FL} . Two early days of block 1 ($n = 46$ cells, 4 mice), two late days of block 1 ($n = 72$ cells, 5 mice), two days around block switch ($n = 86$, 5 mice), two early days of block 2 ($n = 97$ cells, 5 mice), and two late days of block 2 ($n = 103$ cells, 5 mice). Wilcoxon matched-pairs signed rank test results: early 1-2 (block 1) $*p < 0.001$. All other matched day pairs' Wilcoxon matched-pairs signed rank test: $p > 0.05$.

(c) Pf_{FL} population distribution of positively (white), negatively (gray), and non-modulated (dark gray) cells on all matched day pairs. Fisher's exact test (performed on cell counts): E1-E2 (block 1) $*p < 0.05$, E1-E2 (block 2) $*p < 0.05$. All other matched day pairs' Fisher's exact tests: $p > 0.05$.

(d) Same as in (c) for VAL_{FL} . Fisher's exact test (performed on cell counts) for all matched day pairs: $p > 0.05$.

(e) Neural activity of Pf_{FL} around movement start projected into top 3 PC space for late day of block 1 (cyan trace) and early day for block 2 (black trace). -2 seconds before movement onset (green circle), movement start (blue circle), and +2 seconds after movement start (red circle).

(f) Same as (d), for VAL_{FL} matched neuronal populations.

(g) For Pf_{FL} , average alignment of subspaces occupied within (open circle with dot) and across (filled circle) days with matched cells from two early days (E) in each block (black, average over block 1 and 2), two late days (L) in each block (magenta, average over block 1 and 2), and the block switch day from late block 1 to early block 2 (L/E, blue). For Pf_{FL} , paired t-test within vs across; E, $*p < 0.05$; L, $p > 0.05$; L/E, $*p < 0.005$.

(h) Same as (g), for VAL_{FL} matched cell populations. Paired t-test within vs across; E, $*p < 0.001$; L, $*p < 0.01$; L/E, $*p < 0.01$.

See also Extended Data Figs. 6-8

2. It is a bit worrying that only ~35% of Pf and ~15% of VAL were lesioned for the loss of function experiments. On the one hand, it is maybe not surprising that such small lesions, especially in VAL, did not lead to strong behavior deficits; on the other hand, it is surprising that lesioning of even a very small portion of Pf had an effect on motor learning. It appears that even lesioning 5-10% of Pf significantly reduces the hit rate after learning (Fig. S6E). Are these lesions affecting a specific part of Pf that is particularly important for this task? A better characterization of the lesion in Pf and VAL might aid with interpreting these results.

This is an important point for us to clarify. The size of lesions in our experiments were quite small by design to target thalamic nuclei sub-regions that could be more important/sensitive in our task in particular. Specifically, we targeted the forelimb related Pf and VAL for the lesions, which we defined by retrogradely tracing from the forelimb-related striatum and the caudal forelimb area of the motor cortex to map the forelimb-related Pf (Pf_{FL}) and VAL (VAL_{FL}), respectively (see Extended Data Fig. 2). **We have added additional quantifications of these forelimb related areas (Extended Data Fig. 10f-g)** where we show the fraction of the lesioned volume that contained Pf_{FL} and total Pf (or VAL, respectively). These data show that the majority of the lesion volumes were within Pf and VAL or their related forelimb areas. However, when we looked closer into the Pf_{FL} and VAL_{FL} areas, we found that a few animals had less than 10% of these forelimb related areas lesioned. **We have excluded these animals from the study.** However, as our lesion volumes do not span the entirety of Pf or VAL, our results indicate that disrupting a smaller population of cells within these nuclei is enough to disrupt the learning circuit, which may suggest that a small effect locally in thalamus can have a large effect in downstream areas such as striatum and cortex. This is supported by the correlation analysis in Extended Data Fig. 10 h-i.

Extended Data Fig. 10. Behavioral training of pre-learning Pf and VAL lesioned mice in spatial target task

(a) Hit ratio (hits / all attempts) of each animal in pre-learning lesion experiment. High performance criterion (65% hit ratio) indicated in dotted black line. Pf_{pre-lesion} (purple), n = 8; Pf_{pre-control} (black lines), n = 10.

(b) Same as (a) but for VAL pre-learning lesion experiment. VAL_{pre-lesion} (green lines), n = 7; Pf_{pre-control} (black lines), n = 10.

(c) Engagement (# of attempts) on the first day of block 1 and block 2. Pf_{pre-lesion} vs Pf_{pre-control}, unpaired t-test; Block 1: $t(18) = 0.37, p > 0.05$, Block 2: $t(18) = 1, p > 0.05$. VAL_{pre-lesion} vs VAL_{pre-control}, unpaired t-test, Block 1: $t(16) = 0.52, p > 0.05$, Block 2: $t(16) = 0.13, p > 0.05$.

(d) Representative 3D reconstruction of lesions areas (green-volume) of bilateral Pf (top – red volume) and VAL (bottom – red volume) lesion. Made using BrainJ outputs and BrainRender for visualization.

(e) Total lesion volume (mm³) for Pf_{pre-lesion} (purple) and VAL_{pre-lesion} (green) mice.

(f) Relative share of lesioned volume in thalamic nuclei for Pf_{pre-lesion} mice. Relative share of lesion in forelimb-related Pf and total Pf (purple), other thalamic areas (gray).

(g) Same as in (f) but for VAL_{pre-lesion} mice. Relative percent of lesion in forelimb-related VAL and total VAL (green), other thalamic areas (gray)

(h) Percent of Pf lesioned in Pf_{pre-lesion} mice versus hit ratio on the last day of block 1 (hits/all attempts). n=10 mice. Fitted regression line (purple line) shows there is no significant correlation between percent of Pf_{FL} lesioned and hit ratio. Spearman correlation; Pf_{FL}, $r = 0.15, p > 0.05$.

(i) Same as (h), but for VAL_{pre-lesion} lesion cohort, n=7. Fitted regression line (green) shows there is no significant correlation between percent VAL_{FL} lesioned and hit ratio. Spearman correlation; VAL_{FL}, $r = -$

0.13, $p > 0.05$. Fitted regression line (green) shows there is no significant correlation between percent VAL_{FL} lesioned and hit ratio.

(e-g) Mean \pm SEM shown. Pf_{pre-lesion} $n=8$ mice, VAL_{pre-lesion} $n=7$ mice.

(f and g) Acronyms: MD: mediodorsal nucleus; PCN: paracentral nucleus; CL: central lateral; PO: posterior complex of the thalamus; LP: lateral posterior nucleus; VM: ventral medial nucleus; AV: Anteroventral nucleus; AMv: Anteromedial nucleus, ventral part; AMd: Anteromedial nucleus, dorsal part.

- 3. Statistical analysis for figure 3C and S5C is lacking. It appears that the fraction of newly negatively modulated cells has the most drastic difference between two late days vs. a late block 1 and an early block 2 day for both Pf and VAL neurons. Only mentioning the fraction of newly positively modulated cells for Pf and only newly negatively modulated cells for VAL seems arbitrary. Are these the only statistically significant differences between late/late and late/early?**

We apologize for the missing values of statistical tests, which has led to the warranted concern in this comment. All statistical claims mentioned in the text were based on specific statistical tests, but were omitted from the legends. **Figure 3 has changed significantly after additional revision analysis (mainly adding new days for cell matching)**. For these new matched cell pairs, we analyzed the distribution of cells that were positively, negatively, and non-modulated across all matched day pairs. We found that the distribution of cell responses changed significantly for Pf early in block 1 and block 2, but was stable late in both blocks (Fig. 3c). We used a Fisher's exact test to quantify this statistical difference on the raw counts of each response type, while the Figure shows relative fractions of cells. For VAL, there was no significant change in response distributions across any paired day (Fig. 3d).

Figure 3: Low dimensional population activity alignment changes between early and late learning

(a) Pf_{FL} trial-averaged fluorescence aligned to movement start for matched cells on two early days of block 1 (early 1 – black, early 2 – blue, $n = 113$ cells, 5 mice), two late days of block 1 (late 1 – pink, late 2 – cyan, $n = 114$ cells, 5 mice), two days around block switch (late 2 – cyan, early 1 – black, $n = 116$ cells, 5 mice), two early days of block 2 ($n = 67$ cells, 4 mice), and two late days of block 2 ($n = 49$ cells, 4 mice). Wilcoxon matched-pairs signed rank test results: early 1-2 (block 1) $*p < 0.0001$, late 1-2 (block 1) $p > 0.05$, late 2 (block 1) – early 1 (block 2) $*p < 0.0001$, early 1-2 (block 2) $*p < 0.0001$, late 1-2 (block 2) $p > 0.05$. Data shown as mean \pm SEM (thick colored line with shaded bounds).

(b) Same as in (a) for VAL_{FL} . Two early days of block 1 ($n = 46$ cells, 4 mice), two late days of block 1 ($n = 72$ cells, 5 mice), two days around block switch ($n = 86$, 5 mice), two early days of block 2 ($n = 97$ cells, 5 mice), and two late days of block 2 ($n = 103$ cells, 5 mice). Wilcoxon matched-pairs signed rank test results: early 1-2 (block 1) $*p < 0.001$. All other matched day pairs' Wilcoxon matched-pairs signed rank test: $p > 0.05$.

(c) Pf_{FL} population distribution of positively (white), negatively (gray), and non-modulated (dark gray) cells on all matched day pairs. Fisher's exact test (performed on cell counts): E1-E2 (block 1) $*p < 0.05$, E1-E2 (block 2) $*p < 0.05$. All other matched day pairs' Fisher's exact tests: $p > 0.05$.

(d) Same as in (c) for VAL_{FL} . Fisher's exact test (performed on cell counts) for all matched day pairs: $p > 0.05$.

(e) Neural activity of Pf_{FL} around movement start projected into top 3 PC space for late day of block 1 (cyan trace) and early day for block 2 (black trace). -2 seconds before movement onset (green circle), movement start (blue circle), and +2 seconds after movement start (red circle).

(f) Same as (d), for VAL_{FL} matched neuronal populations.

(g) For Pf_{FL} , average alignment of subspaces occupied within (open circle with dot) and across (filled circle) days with matched cells from two early days (E) in each block (black, average over block 1 and 2), two late days (L) in each block (magenta, average over block 1 and 2), and the block switch day from late block 1 to early block 2 (L/E, blue). For Pf_{FL} , paired t-test within vs across; E, $*p < 0.05$; L, $p > 0.05$; L/E, $*p < 0.005$.

(h) Same as (g), for VAL_{FL} matched cell populations. Paired t-test within vs across; E, $*p < 0.001$; L, $*p < 0.01$; L/E, $*p < 0.01$.

See also Extended Data Figs. 6-8

4. Does the activity amplitude of neurons that show the same type of modulation across late block 1 and early block 2 change between those to time points? Or is the increase in activity on early block 2 days in Pf primarily driven by newly positively modulated cells?

The amplitude of neurons with positive modulation in Pf does increase at the block switch (Extended Data Figs. 4c and 6c) and there are also more cells that become positively modulated on early learning days (Fig. 3c). With this, there seems to be a contribution from both change in modulation (amplitude) and increase in positively modulated cells.

From Extended Data Fig. 4c. Classification of thalamic cells by responses during movement

c) Change in response for Pf positively (left), negatively (middle), and not modulated (right) cells on early block 1 (solid black line), late block 1 (cyan line), and early block 2 (dashed black line). Kruskal-Wallis test, positive population activity at start: $*p < 0.0001$; Dunn's multiple comparisons test, early block 1 vs late block 1: $*p < 0.0001$, early block 2 vs late block 1: $*p = 0.0001$. Kruskal-Wallis test, negative population activity: $*p < 0.05$; Dunn's multiple comparisons test, early block 1 vs late block 1: $*p < 0.05$, early block 2 vs late block 1: $p > 0.05$. Kruskal-Wallis test, not modulated population activity: $p > 0.05$; Dunn's multiple comparisons test not significant for all pairs.

From Extended Data Fig. 6c. Changing matched cellular responses during movement over multiple days

(c) For Pf_{FL} matched cell populations, trial-averaged fluorescence aligned to movement start for positively modulated cells. Mann-Whitney test, fluorescence at time of movement start: early 1 vs early 2 (block 1) $p > 0.05$, late 1 vs late 2 (block 1) $p > 0.05$, late 2 (block 1) vs early 1 (block 2) $*p < 0.0001$, early 1 vs early 2 (block 2) $*p < 0.05$, late 1 vs late 2 (block 2) $p > 0.05$.

From Figure 3c: Low dimensional population activity alignment changes between early and late learning

(c) Pf_{FL} population distribution of positively (white), negatively (gray), and non-modulated (dark gray) cells on all matched day pairs. Fisher's exact test (performed on cell counts): E1-E2 (block 1) $*p < 0.05$, E1-E2 (block 2) $*p < 0.05$. All other matched day pairs' Fisher's exact tests: $p > 0.05$.

5. The authors focus their quantification in Fig. 2l on movement activity, but how does overall activity at different phases of the task (pre-movement, movement, post-movement, reward) change with training. There appears to be quite a strong change in the activity at those pre and post movement phases.

This is an interesting point, as we agree that there is a lot to explore in our dataset. Our initial analysis of the neural data agnostically analyzed during which phase (pre-movement, during movement, or after hit) the largest fraction of cells were significantly modulated compared to a baseline period (Fig. 2f). On average only around 20% of cells were modulated during the 'pre-movement' phase, and 12% during the 'after hit' phases, but a very high fraction of cells was modulated during the movement phase itself. We therefore focused our further analysis on the movement phase. While there is a change in activity with learning across the whole activity trace shown in Fig. 4g/h, we find that the peak activity appears consistently shortly after the movement onset. This

further supports that the main neuronal activity is unfolding during the ongoing movement.

The post-movement time window includes reward delivery and consumption, which in itself is quite interesting, but was conceptually outside of the scope of our project since we are not focused on reward processing. Furthermore, while we found some cells that are selectively modulated by reward, they are few in number, making it difficult to interrogate them robustly.

We have added further analysis investigating pre-movement time window. To test whether behaviorally relevant information is encoded in pre-movement activity, **we added additional analysis specifically comparing pre- and post-movement onset.** We performed new regression analysis using activity pre or post-movement onset from Pf and VAL to predict the initial reach direction, and compared the performance of each model (Figure 4f-g). We found that the model using Pf activity pre-onset performed significantly poorer than the model using activity during movement to predict the initial movement direction of a given trial.

These results suggest that the thalamic activity during the movement is most relevant for the behavior investigated here.

From Figure 2e-h. Pf_{FL} and VAL_{FL} neural populations are most active during reaches early in learning

(e) Schematic of the three time windows of a reach; pre-movement (white), during movement (yellow), after hit (gray).

(f) Top: proportion of significantly modulated cells on early day of block 1 during the time windows depicted in (e) for Pf_{FL} , $n = 5$ mice. Pf_{FL} : Repeated measures one-way ANOVA, $F(2,7) = 31$, $*p < 0.005$; Šídák's multiple comparisons: pre-movement vs during movement: $*p < 0.01$, during movement vs after hit: $*p < 0.01$. Bottom: same for VAL_{FL} , $n = 5$ mice. Repeated measures one-way ANOVA: $F(1,6) = 15$, $*p < 0.01$; Šídák's multiple comparisons: pre-movement vs during movement: $*p < 0.01$, during movement vs after hit: $*p < 0.05$. Data shown as mean \pm SEM.

(g) Top: trial-average z-scored fluorescence of all cells on 3 sessions from block 1; early (black), middle (green), late (cyan) for Pf_{FL} . Kruskal-Wallis test, population activity at start: $*p < 0.0001$; Dunn's multiple comparisons test, early vs mid: $*p < 0.005$, early vs late: $*p < 0.0001$, mid vs late: $p > 0.05$. Bottom: same

as top for VAL_{FL} data. Kruskal-Wallis test, population activity at start: * $p < 0.001$; Dunn's multiple comparisons test, early vs mid: * $p < 0.005$, early vs late: * $p < 0.005$, mid vs late: * $p < 0.0001$. Data shown as mean \pm SEM.

(h) Same as (g) for block 2; Top: Pf_{FL} Kruskal-Wallis test, population activity at start: * $p < 0.0001$; Dunn's multiple comparisons test, early vs mid: * $p < 0.0001$, early vs late: * $p < 0.0001$, mid vs late: $p > 0.05$. Bottom: same as top for VAL_{FL} data. Kruskal-Wallis test, population activity at start: $p > 0.05$. Data shown as mean \pm SEM.

From Figure 4f-g. Pf_{FL} neural activity encodes reach direction early, but not late, in learning

f) Averaged coefficients of determination (R^2 s) of model predictions for Pf_{FL} using neural activity pre (black) and post (purple) onset of movement. 2-way repeated measures ANOVA: time from onset effect, $F(1,4) = 62$: * $p < 0.005$. Šídák's multiple comparisons test: pre vs post (early), * $p < 0.001$, pre vs post (early-mid), * $p < 0.01$, pre vs post (mid), * $p < 0.01$, pre vs post (mid-late), $p > 0.05$, pre vs post (late), * $p = 0.0001$.

(g) Same as in (f), for VAL_{FL} neural data pre (black) and post (green) movement onset. 2-way repeated measures ANOVA: time from onset effect, $p > 0.05$.

I have a few additional minor concerns.

1. Is the early block 2 training day used for the different analyses in Fig. 3 the "first" day of training with the new target? The text is not entirely clear.

Yes, the early day is the first day of the new target in Fig 3. We have also added a clarification in the text.

Page 5, lines 163-166

To investigate this, we repeated the previous analysis using matched cells from two early days (early 1 and early 2) and two late days (late 1 and late 2) per block, as well as cells that were matched across the late day of block 1 and the early day of block 2 (which is the first day of block 2) (Fig. 3a-b).

Also, Fig. 3 H and I are labeled as comparison "within day" and "across day"; are those comparisons indeed made within the same day or across days either within late block 1 or across late block 1 and early block 2? It should be across days.

The analysis performed in this Figure and the new Figure 3g-h was done by splitting the data from a session into two datasets. This allows us to perform the alignment

analysis twice for each session. First, within its own session (this tells us how stable the alignment within a session is, and is represented by open circles). It represents a 'baseline' of how stable neural activity is within a given session. Second, we compare across two sessions using matched cells (this tells us how aligned the activity is across sessions, and is represented by filled circles). This method allows us to determine how aligned the neural activity is between two sessions, by comparing it to the alignment of the same cells within a single session. If the alignment across sessions is significantly lower than within a session, we can conclude that the neural activity is unstable across the two analyzed sessions (pair-wise comparisons of Fig. 3g-h).

From Figure 3g-h: Low dimensional population activity alignment changes between early and late learning

(g) For Pf_{FL}, average alignment of subspaces occupied within (open circle with dot) and across (filled circle) days with matched cells from two early days (E) in each block (black, average over block 1 and 2), two late days (L) in each block (magenta, average over block 1 and 2), and the block switch day from late block 1 to early block 2 (L/E, blue). For Pf_{FL}, paired t-test within vs across; E, * $p < 0.05$; L, $p > 0.05$; L/E, * $p < 0.005$.

(h) Same as (g), for VAL_{FL} matched cell populations. Paired t-test within vs across; E, * $p < 0.001$; L, * $p < 0.01$; L/E, * $p < 0.01$.

Please also double check the statistics presented for Figs. 3H and I (lines 228-239, are those the p-values for within vs across comparisons or L vs E and L vs L?).

We apologize again for the missing values of statistical tests. We have added all the relevant analysis for both panels (see just above).

2. It would be helpful to show the behavior trajectories and quantifications on the chosen days for the cross-day analyses in Fig. 3 and S5

The days in the cross-day analysis in Fig. 3 and Extended Data Fig. 7 are the same as the data in Figure 1. We did not use any sessions for the neural activity analysis that was not included in Figure 1.

- For the reach direction encoding analysis the authors subsample "trials of all days to match the lowest reach direction variability". While I understand the need for this, I wonder what the implications are. What are the properties of these trials chosen by subsampling? It would be helpful to show their hit rate etc.

We thank the reviewer for the suggestion and have included a new Extended Data Fig. 9 called "Subsampling trials for angle variance matching" showing variance and hit ratio before and after sub-sampling trials. Please see this new figure below. The behavioral changes we see across learning are preserved even when we subsample the trials for directional variance matching.

Extended Data Fig. 9. Subsampling trials for angular variance matching

- (a) Representative polar plot of initial vector direction distribution before (left) and after (right) variability matching.
- (b) Number of attempts in 5 training session per block before (left, blue) and after (right, green) variability matching. Red indicates excluded data with less than 50 trials in a session left after variance matching.
- (c) Directional variance of initial direction of movement before (left, blue) and after (right, green).
- (d) Hit ratio of animals across two training blocks before (left, blue) and after (right, green).
- (b-c) Mean \pm SEM is shown for each day, single animals are shown in each point ($n = 10$ mice).

4. Can Pf neural activity predict reach direction in early training with the second target?

Yes, the Pf neural activity can predict reach direction in early training within the second target. **We have updated and expanded our initial analysis from Figure 4 to better show this.** In Fig. 4e we added additional days over both blocks for our direction regression analysis and plot the block averaged results, which indicate the Pf neural activity predicts reach direction early in both block 1 and block 2.

Figure 4. Pf_{FL} neural activity encodes reach direction early, but not late, in learning

- (a) Predicted initial vector direction of combined held-out test data from 10 models trained on Pf_{FL} neural activity using different train/test splits, plotted against the corresponding true initial vector direction. Models trained on an early session in block 1.
- (b) Same as (a), for VAL_{FL} animal.
- (c) Circular histogram showing the true (left) and predicted (right) initial vector directions from Pf_{FL} animal data in (a).
- (d) Circular histogram showing the true (left) and predicted (right) initial vector directions from VAL_{FL} animal data in (b).

(e) Averaged coefficients of determination (R^2 s) of model predictions for Pf_{FL} (purple), VAL_{FL} (green), and models trained on shuffled vector data (gray) for 5 days of training in each block (averaged over two blocks). 2-way repeated measures ANOVA: day effect, $F(4,32) = 4$: $*p < 0.05$, region effect, $F(1,8) = 17$: $*p < 0.005$. Šidák's multiple comparisons test: Pf_{FL} early-mid vs VAL_{FL} early-mid, $*p < 0.05$. Pf_{FL} mid vs VAL_{FL} mid, $*p < 0.05$. Pf_{FL} late vs VAL_{FL} late, $*p < 0.05$. For all other day comparisons, $p > 0.05$. For shuffled data: 2-way repeated measures ANOVA, day and region effects, $p > 0.05$.

(f) Averaged coefficients of determination (R^2 s) of model predictions for Pf_{FL} using neural activity pre (black) and post (purple) onset of movement. 2-way repeated measures ANOVA: time from onset effect, $F(1,4) = 62$: $*p < 0.005$. Šidák's multiple comparisons test: pre vs post (early), $*p < 0.001$, pre vs post (early-mid), $*p < 0.01$, pre vs post (mid), $*p < 0.01$, pre vs post (mid-late), $p > 0.05$, pre vs post (late), $*p = 0.0001$.

(g) Same as in (f), for VAL_{FL} neural data pre (black) and post (green) movement onset. 2-way repeated measures ANOVA: time from onset effect, $p > 0.05$.

(h) Averaged coefficients of determination (R^2 s) of multivariate model predictions for Pf_{FL} (purple), VAL_{FL} (green), early day block average. 2-way repeated measures ANOVA for Pf_{FL} : Model effect, $F(4,32) = 19$: $*p < 0.0001$. Šidák's multiple comparisons test for Pf_{FL} : direction vs full model, $*p < 0.05$. Direction vs speed/distance, $*p < 0.0001$. Full model vs speed/distance, $*p < 0.05$. Direction/distance vs speed/distance, $*p < 0.005$. Direction/speed vs speed/distance, $*p < 0.005$. For all other comparisons, $p > 0.05$. Šidák's multiple comparisons test for VAL_{FL} : Direction vs speed/distance, $*p = 0.0001$. Direction/distance vs speed/distance, $*p < 0.01$. Direction/speed vs speed/distance, $*p < 0.005$. For all other comparisons, $p > 0.05$.

See also Extended Data Figs. 8 and 9

5. It would be helpful to show the outlines of Pf and VAL in the immunostaining images of Fig 5 B-E, so that the reader can gauge the lesion coverage.

We have added the outlines of Pf and VAL to this panel as suggested. They are now shown in the dashed white-lines in Fig 5b-e.

From Figure 5a-e. Ablation of Pf_{FL} but not VAL_{FL} cells before learning impairs the refinement of initial reach direction

(a) Experimental design showing the timeline for surgeries and STT training, and coronal ABA sections showing the injection targets for Pf_{FL} and VAL_{FL} bilateral lesions.

(b) Representative coronal section of $VGlut2-Cre$ mouse injected with AAV expressing $taCasp3$ - NeuroTrace (yellow) targeted to Pf_{FL} . Dashed-line indicating Pf. High-magnification view from the inset in box.

(c) Representative coronal section of WT-littermate mouse injected with AAV expressing $taCasp3$ - NeuroTrace (yellow) targeted to Pf_{FL} . Dashed-line indicating Pf. High-magnification view from the inset in box.

(d-e) Same as (b-c), for AAV injection targeting VAL_{FL} . Dashed-line indicating VAL.

Reviewer #3 (Remarks to the Author):

The thalamus is a central hub of the brain, interlinking distributed motor areas including the motor cortex, basal ganglia, and cerebellum. These motor areas likely have distinct roles during motor learning due to many factors, including the unique teaching signals they receive. Notably, different thalamic nuclei are connected with various parts of motor networks, making the thalamus an interesting brain area to monitor and manipulate activity to infer the functional roles of these networks.

Sibener et al. focused on the Pf and VAL nuclei, which receive input from the basal ganglia (BG) and deep cerebellar nuclei (DCN), respectively. Additionally, Pf projects back to the BG while VAL mainly projects back to the motor cortex (MCx). The study investigated how the activity of Pf and VAL changes during learning and tested their causality in learning through lesion experiments. For both recording and lesion experiments, significant differences between Pf and VAL were discovered, which is intriguing and important for the field. However, there are several concerns that need to be addressed before publication, as listed below.

Major

1) Movement changes over training: As described in detail in Fig. 1, movement changes over the course of training, yet this has not been considered for major parts of the activity analysis (except in Fig. 4). Trial average activity could be high during learning because animals explore more speeds and locations. For example, these neurons might have a "receptive field" for particular locations or speeds of arm movement. If so, when mice explore various trajectories, in some trials, mice move within this receptive field, causing average activity to become higher.

While trial averages are a good starting point, more sophisticated analyses considering differences in movement across trials are required to determine whether the activity in these thalamic nuclei changes independently of movement patterns during learning. Comparing activity in trials with similar trajectories and speeds across learning stages is one way to approach this. Alternatively, a regression model, like the one used in Fig. 4 (but not limited to initial movement direction), could be useful. A similar consideration of trial-by-trial movement is required for the PCA analysis in Fig. 3.

We thank the reviewer for this important comment. We have addressed this concern in three distinct ways.

- To take into consideration the movement changes in training, **we have re-analyzed and plotted trial-averaged responses over each block for Pf and VAL populations, by subsampling trajectories to a matched directional variance (see below)**. When subsampling the behavior to control for the change in directional variability over time, we still find the same changes in Pf and VAL neural population responses over block 1 and 2, namely that Pf neural populations decrease the magnitude of responses over training of both blocks (see below figure, all trials vs subsampled trials). As the subsampling did not yield a different result, we did not replace the original figure panels in the manuscript (Figure 2g-h), but can do so at the reviewers request.

Comparing trail-averaged activity for all trials and directional variance matched trials

(a) **Top left:** trial-average z-scored fluorescence of all cells on 3 sessions from block 1; early (black), middle (green), late (cyan) for Pf_{FL} . Kruskal-Wallis test, population activity at start: $*p < 0.0001$; Dunn's multiple comparisons test, early vs mid : $p > 0.05$, early vs late: $*p < 0.0001$, mid vs late: $p > 0.05$. **Bottom left:** same as top for VAL_{FL} data. Kruskal-Wallis test, population activity at start: $*p < 0.001$; Dunn's multiple comparisons test, early vs mid : $*p < 0.005$, early vs late: $*p < 0.005$, mid vs late: $*p < 0.0001$. **Top right:** same as in top left for block 2. Pf_{FL} Kruskal-Wallis test, population activity at start: $*p < 0.0001$; Dunn's multiple comparisons test, early vs mid : $*p < 0.0001$, early vs late: $*p < 0.0001$, mid vs late: $p > 0.05$. **Bottom right:** same as bottom left for VAL_{FL} block 2 data. Kruskal-Wallis test, population activity at start: $p > 0.05$. Data shown as mean \pm SEM.

(b) Same as in (a) using trials subsampled to matched directional variance. **Top left:** Kruskal-Wallis test Pf_{FL} block 1, population activity at start: $*p < 0.05$; Dunn's multiple comparisons test, early vs mid : $*p < 0.05$, early vs late: $p > 0.05$, mid vs late: $p > 0.05$. **Bottom left:** same as top for VAL_{FL} data. Kruskal-Wallis test, population activity at start: $*p < 0.0005$; Dunn's multiple comparisons test, early vs mid : $p > 0.05$, early vs late: $*p < 0.005$, mid vs late: $*p < 0.0005$. **Top right:** same as in top left for Pf_{FL} block 2. Pf_{FL} Kruskal-Wallis test, population activity at start: $*p < 0.0001$; Dunn's multiple comparisons test, early vs mid : $*p < 0.05$, early vs late: $*p < 0.0001$, mid vs late: $p > 0.05$. **Bottom right:** same as bottom left for VAL_{FL} block 2 data. Kruskal-Wallis test, population activity at start: $p > 0.05$.

- Following the suggestion of the reviewer, **we expanded on the regression analysis to investigate the behavior parameters of speed and distance, in addition to direction**. Specifically, we trained Ridge regression models to predict the initial direction of movement (Fig. 4e), the maximum speed (Extended Data Fig. 8i), or maximum distance for each trial (Extended Data Fig. 8j), separately matching the trials for variance in each behavioral variable. Neither maximum speed nor

maximum distance was encoded strongly in the neural data. We further trained a Full Model where all three behavioral variables (direction, speed, and distance) were used (Fig. 4h, Extended Data Fig. 8e). Lastly, we trained additional models where one predicted variable (i.e. direction, speed, or distance) was removed from the Full Model (Figure 4h, Extended Data Figure 8f-h). By comparing the R^2 s of these models on early days of the blocks, we found that initial direction was best decoded. The full model, which included direction, speed, and distance, performed significantly worse than the model using direction information alone, suggesting that other behavioral variables besides direction were preventing the regression model from being fit well to the data. This new analysis provides strong evidence that direction is specifically encoded, particularly in Pf activity.

Figure 4. Pf_{FL} neural activity encodes reach direction early, but not late, in learning

(a) Predicted initial vector direction of combined held-out test data from 10 models trained on Pf_{FL} neural activity using different train/test splits, plotted against the corresponding true initial vector direction. Models trained on an early session in block 1.

(b) Same as (a), for VAL_{FL} animal.

(c) Circular histogram showing the true (left) and predicted (right) initial vector directions from Pf_{FL} animal data in (a).

(d) Circular histogram showing the true (left) and predicted (right) initial vector directions from VAL_{FL} animal data in (b).

(e) Averaged coefficients of determination (R^2 s) of model predictions for Pf_{FL} (purple), VAL_{FL} (green), and models trained on shuffled vector data (gray) for 5 days of training in each block (averaged over two blocks). 2-way repeated measures ANOVA: day effect, $F(4,32) = 4$; $*p < 0.05$, region effect, $F(1,8) = 17$; $*p < 0.005$.

Sidak's multiple comparisons test: Pf_{FL} early-mid vs VAL_{FL} early-mid, $*p < 0.05$. Pf_{FL} mid vs VAL_{FL} mid, $*p < 0.05$. Pf_{FL} late vs VAL_{FL} late, $*p < 0.05$. For all other day comparisons, $p > 0.05$. For shuffled data: 2-way repeated measures ANOVA, day and region effects, $p > 0.05$.

(f) Averaged coefficients of determination (R^2 s) of model predictions for Pf_{FL} using neural activity pre (black) and post (purple) onset of movement. 2-way repeated measures ANOVA: time from onset effect, $F(1,4) = 62$; $*p < 0.005$.

Sidak's multiple comparisons test: pre vs post (early), $*p < 0.001$, pre vs post (early-mid), $*p < 0.01$, pre vs post (mid), $*p < 0.01$, pre vs post (mid-late), $p > 0.05$, pre vs post (late), $*p = 0.0001$.

(g) Same as in (f), for VAL_{FL} neural data pre (black) and post (green) movement onset. 2-way repeated measures ANOVA: time from onset effect, $p > 0.05$.

(h) Averaged coefficients of determination (R^2 s) of multivariate model predictions for Pf_{FL} (purple), VAL_{FL} (green), early day block average. 2-way repeated measures ANOVA for Pf_{FL}: Model effect, $F(4,32) = 19$: $*p < 0.0001$. Šídák's multiple comparisons test for Pf_{FL}: direction vs full model, $*p < 0.05$. Direction vs speed/distance, $*p < 0.0001$. Full model vs speed/distance, $*p < 0.05$. Direction/distance vs speed/distance, $*p < 0.005$. Direction/speed vs speed/distance, $*p < 0.005$. For all other comparisons, $p > 0.05$. Šídák's multiple comparisons test for VAL_{FL}: Direction vs speed/distance, $*p = 0.0001$. Direction/distance vs speed/distance, $*p < 0.01$. Direction/speed vs speed/distance, $*p < 0.005$. For all other comparisons, $p > 0.05$.

See also Extended Data Figs. 8 and 9

From Extended Data Fig. 8. Direction tuning and Ridge regressions for reaching variables

(e) Averaged coefficients of determination (R^2 s) of full model (direction, distance, and speed) predictions for Pf_{FL} (purple), VAL_{FL} (green) for 5 days of training in each block (averaged over two blocks). 2-way repeated measures ANOVA: day effect, $F(4,32) = 4$: $*p < 0.05$, region effect, $F(1,8) = 14$: $*p < 0.01$. Šídák's multiple comparisons test: Pf_{FL} early vs VAL_{FL} early, $*p < 0.05$. Pf_{FL} early-mid vs VAL_{FL} early-mid, $*p < 0.05$. Pf_{FL} mid vs VAL_{FL} mid, $*p < 0.05$. For all other comparisons, $p > 0.05$.

(f) Same as in (e) for the model predicting trial initial direction and total distance. 2-way repeated measures ANOVA: day effect, $F(4,32) = 4$: $*p < 0.05$, region effect, $F(1,8) = 15$: $*p < 0.01$. Šídák's multiple comparisons test: Pf_{FL} mid vs VAL_{FL} mid, $*p < 0.05$. For all other comparisons, $p > 0.05$.

(g) Same as in (e) for model predicting trial maximum speed and maximum distance. 2-way repeated measures ANOVA: day effect, $F(4,32) = 1$: $p > 0.05$, region effect, $F(1,8) = 3$: $p > 0.05$.

(h) Same as in (e) for model predicting trial initial direction and maximum speed. 2-way repeated measures ANOVA: day effect, $F(4,32) = 3$: $*p < 0.05$, region effect, $F(1,8) = 13$: $*p < 0.01$. Šídák's multiple comparisons test: Pf_{FL} early-mid vs VAL_{FL} early-mid, $*p < 0.05$. Pf_{FL} mid vs VAL_{FL} mid, $*p < 0.05$. For all other comparisons, $p > 0.05$.

(i) Averaged coefficients of determination (R^2 s) of maximum speed model predictions for Pf_{FL} (purple), VAL_{FL} (green), and models trained on shuffled vector data (gray). Mixed-effects analysis: day effect, $F(4,31) = 1$: $p > 0.05$, region effect $F(1,8) = 2$, $p > 0.05$. For shuffled data: Mixed-effects analysis, day and region effects, $p > 0.05$.

(j) Same as in (i) for model predicting trial maximum distance. Mixed-effect analysis: day effect, $F(4,31) = 0.5$, $p > 0.05$, region effect $F(1,8) = 1$, $p > 0.05$. For shuffled data: Mixed-effects analysis, day and region effects, $p > 0.05$.

(a-b and e-j) Mean \pm SEM is shown for each day, single animals are shown in each point (Pf_{FL}, $n = 5$ mice; VAL_{FL}, $n = 5$ mice).

We followed the suggestions of the reviewer to investigate the 'receptive fields' or direction tuning of thalamic neurons, given that our regression models indicated direction was decoded from Pf neural activity and head-direction cells have been reported in other thalamic nuclei (Taube, 1995; Cho and Sharp, 2001). We have added a new analysis where we identify cells in Pf and VAL with reach direction tuning

(Extended Data Fig. 8a-c). We found that the proportion of cells with direction tuning was decreased over in learning in Pf, and Pf had significantly more direction tuned cells than VAL, in line with our regression analysis. We also investigated whether cells that were matched across days of training would maintain their tuning properties, but this analysis was confounded by the limited number of matched cells that were also tuned on a given day. However, our limited analysis showed that only a small proportion of cells were stably tuned for a specific direction, and larger proportions were losing or changing their tuning across days. These findings suggest that the population activity changes throughout learning are **not** driven by stably tuned cells that get differentially activated as the behavior changes.

From Extended Data Fig. 8. Direction tuning and Ridge regressions for reaching variables

(a) Representative examples of cells with directional tuning.

(b) Proportion of direction tuned cells in Pf_{FL} (purple) and VAL_{FL} (green). Data is block averaged for 5 selected days. 2-way repeated measures ANOVA, day effect, $F(4,32) = 4$: * $p < 0.01$, region effect, $F(1,8) = 9$: * $p < 0.05$, Šidák's multiple comparisons test: Pf_{FL} early vs VAL_{FL} early, * $p < 0.05$, Pf_{FL} mid vs VAL_{FL} mid, * $p < 0.05$. For all other day comparisons, $p > 0.05$.

(c) Same as in (b), for trials that are matched for directional variance. 2-way repeated measures ANOVA, day effect, $F(4,32) = 5$: * $p < 0.005$, region effect, $F(1,8) = 7$: * $p < 0.05$, Šidák's multiple comparisons test: Pf_{FL} mid vs VAL_{FL} mid, * $p < 0.05$. For all other day comparisons, $p > 0.05$.

(d) Relationship between the initial vector variability and coefficient of determination (R^2 's) from models using all trials on a given day. Linear fit for test (orange), train (blue) and shuffle (gray) models shown. Simple linear regressions, slope different from zero; test (* $p < 0.0001$), train (* $p < 0.0001$), and shuffle (* $p < 0.005$).

2) Fig. 4 analysis: If Pf activity “predicts” reach direction, it would be better to analyze the correlation between reach direction and activity before the onset of movement. Activity after the onset could be a mixture of sensory feedback, efference copy, etc, in addition to activity that drives movement.

We used the word “predict” here in a technical sense related to the regression model that predicts a behavioral variable from neural activity, not in reference to a particular behavioral time point in the future. To clarify this, we have updated the title for Figure 4 to use the word “encodes” rather than “predicts”. However, the reviewer makes an

interesting point that neuronal activity preceding the movement could carry more information about the upcoming reach direction. We have therefore added an additional regression analysis that uses neural activity from Pf or VAL before movement-onset, and compared it to the model that uses neural activity just after movement-onset (Figure 4f-g). By comparing model performances, we found that particularly Pf neural activity better encodes direction during the movement than before movement onset, which was not the case for VAL neural activity. This strongly suggests that Pf relays to its downstream targets information about the ongoing movement, which may also include sensory information, which can be reinforced if it leads to success and aid learning.

Figure 4. Pf_{FL} neural activity encodes reach direction early, but not late, in learning

(a) Predicted initial vector direction of combined held-out test data from 10 models trained on Pf_{FL} neural activity using different train/test splits, plotted against the corresponding true initial vector direction. Models trained on an early session in block 1.

(b) Same as (a), for VAL_{FL} animal.

(c) Circular histogram showing the true (left) and predicted (right) initial vector directions from Pf_{FL} animal data in (a).

(d) Circular histogram showing the true (left) and predicted (right) initial vector directions from VAL_{FL} animal data in (b).

(e) Averaged coefficients of determination (R^2 s) of model predictions for Pf_{FL} (purple), VAL_{FL} (green), and models trained on shuffled vector data (gray) for 5 days of training in each block (averaged over two blocks). 2-way repeated measures ANOVA: day effect, $F(4,32) = 4$: * $p < 0.05$, region effect, $F(1,8) = 17$: * $p < 0.005$. Šidák's multiple comparisons test: Pf_{FL} early-mid vs VAL_{FL} early-mid, * $p < 0.05$. Pf_{FL} mid vs VAL_{FL} mid, * $p < 0.05$. Pf_{FL} late vs VAL_{FL} late, * $p < 0.05$. For all other day comparisons, $p > 0.05$. For shuffled data: 2-way repeated measures ANOVA, day and region effects, $p > 0.05$.

(f) Averaged coefficients of determination (R^2 s) of model predictions for Pf_{FL} using neural activity pre (black) and post (purple) onset of movement. 2-way repeated measures ANOVA: time from onset effect, $F(1,4) = 62$: * $p < 0.005$. Šidák's multiple comparisons test: pre vs post (early), * $p < 0.001$, pre vs post (early-mid), * $p < 0.01$, pre vs post (mid), * $p < 0.01$, pre vs post (mid-late), $p > 0.05$, pre vs post (late), * $p = 0.0001$.

(g) Same as in (f), for VAL_{FL} neural data pre (black) and post (green) movement onset. 2-way repeated measures ANOVA: time from onset effect, $p > 0.05$.

(h) Averaged coefficients of determination (R^2 s) of multivariate model predictions for Pf_{FL} (purple), VAL_{FL} (green), early day block average. 2-way repeated measures ANOVA for Pf_{FL}: Model effect, $F(4,32) = 19$: $*p < 0.0001$. Šídák's multiple comparisons test for Pf_{FL}: direction vs full model, $*p < 0.05$. Direction vs speed/distance, $*p < 0.0001$. Full model vs speed/distance, $*p < 0.05$. Direction/distance vs speed/distance, $*p < 0.005$. Direction/speed vs speed/distance, $*p < 0.005$. For all other comparisons, $p > 0.05$. Šídák's multiple comparisons test for VAL_{FL}: Direction vs speed/distance, $*p = 0.0001$. Direction/distance vs speed/distance, $*p < 0.01$. Direction/speed vs speed/distance, $*p < 0.005$. For all other comparisons, $p > 0.05$. See also Extended Data Figs. 8 and 9

3) Testing necessity during learning vs. expert in Fig. 6: It would be beneficial if the authors had retrained these mice for block 2 to confirm the lesions' effect on learning: then they can make a within animal comparison to conclude that these manipulations affect learning stronger than execution in expert mice. If the authors have performed such experiments, please report them.

AND

4) Thalamic lesion effect on learned action: It would be beneficial if the authors had retrained these mice for block 2 to confirm the lesions' effect on learning. This approach would allow for a within-animal comparison to conclude that these manipulations affect learning more strongly than execution in expert mice. If the authors have performed such experiments, please report them.

We agree that retraining mice on a new target in block 2 would be interesting to further dissect the role of Pf or VAL in learning and performance. Unfortunately, we did not perform such experiments in this initial study. However, we believe that our pre-learning lesions in Pf (Figure 5f) have strongly demonstrated pre-learning lesions leads to a significant impairment in learning of directional reaches. In future studies, it would be worthwhile to conduct an experiment as suggested, to test mice on a new block 2 to further confirm lesion effects on learning.

Minor

1) Fig. 1f-k: These panels need statistics (apologies if I have missed them).

We apologize for the missing values of statistical tests, which has led to the warranted concern in this comment. All statistical claims mentioned in the text were based on specific statistical tests, but were omitted from the legends. We have corrected this mistake and added statistics for all relevant panels in Figure 1f-k (on page 1 of Figure Document, and below).

Figure 1 : Mice explore and refine directional spatial target reaches

(a) Schematic of a head-fixed mouse performing the Spatial Target Task (STT) during imaging.

(b) Left: top view schematic of the Selective Compliance Articulated Robot Arm (SCARA) joystick showing the start position (green circle) and example position of target 1 (blue circle). Right: same as left, example position of target 2 (magenta circle) and previous target 1 (gray dashed area).

(c) Classification of trajectories from self-paced joystick movements. Hit trajectories (yellow line), and miss trajectories (gray line), started with the mouse moving the joystick out of the start position (green circle). Hit trajectories were rewarded upon target hit and ended by motors returning the joystick to the start 750 msec later. Miss trajectories were aborted by motors returning the joystick to the start if the animal let go of the joystick or the attempt timed out (after 7.5 sec).

(d) Schematic showing the phases of the training protocol: during 'pre-training' animals were rewarded first for touching the joystick, then for pushing it in a forward direction. During blocks 1 and 2, attempts that entered targets 1 or 2 were rewarded, respectively.

(e) Left: representative example trajectories on early and late days of block 1 with rewarded target (blue circle) and non-rewarding target (gray circle). Right: same for block 2 with rewarded target (magenta circle) and non-rewarding target (gray circle). Hit trajectories (yellow) that enter the rewarded target and miss trajectories (grey) are plotted in the two-dimensional workspace. Point of entering target and reward delivery is shown as small dark blue circle.

(f) Hit ratio on 5 selected days per block from early training day (E) to late training day (L) and 3 equidistant days in between for each animal. **Mixed-effects model, day in block: $F(3,25) = 40$, $*p < 0.0001$, day x block: $F(3,14) = 4$, $*p < 0.05$. Šidák's multiple comparisons test: E vs L block 1, $*p < 0.005$; E vs L block 2, $*p < 0.0001$.**

(g) Left: representative example heatmap of workspace occupancy of all trajectories of an early and late session (# times a bin is visited). Right: occupancy of workspace (mm^2). **Mixed-effects model, day in**

block: $F(3,23) = 13$, $*p < 0.0001$, mouse: $F(1,9)=6$, $*p < 0.05$. Šídák's multiple comparisons test: E vs L block 1, $*p < 0.05$; E vs L block 2, $*p < 0.005$.

(h) Variability of mean trajectory from all movements (averaged std along the length of the trajectory, mm). Mixed-effects model, day in block: $F(2,20) = 19$, $*p < 0.0001$, mouse: $F(1,9) = 17$, $*p < 0.01$. Šídák's multiple comparisons test: E vs L block 1, $*p < 0.005$; E vs L block 2, $*p < 0.005$.

(i) Variability of initial direction of movement (angular std, a.u). Mixed-effects model, day in block: $F(3,22) = 8$, $*p < 0.01$, mouse: $F(1,9) = 7$, $*p < 0.5$. Šídák's multiple comparisons test: E vs L block 1, $*p < 0.05$; E vs L block 2, $*p < 0.005$.

(j) Peak speed of trajectories (mm/sec). Mixed-effects model, day in block: $F(4, 36) = 4$, $*p < 0.05$, mouse: $F(1,9) = 8$, $*p < 0.05$. Šídák's multiple comparisons test: E vs L block 1, $p > 0.05$; E vs L block 2, $*p < 0.05$.

(k) Average target overshoot (mm) from target entry for all hit trajectories. Mixed-effects model, $F(3,25) = 8$, $*p < 0.001$, mouse: $F(1,9) = 9$, $*p < 0.05$. Šídák's multiple comparisons test: E vs L block 1, $p > 0.05$; E vs L block 2, $*p < 0.05$.

(f-k) Mean \pm SEM is shown in thick color lines and shaded bounds (block 1: blue, block 2: magenta), single animals are shown in gray lines ($n = 10$ mice).

See also Extended Data Fig. 1

2) All figures: First letters should be capitalized.

We have updated all figure legends so that the first letters are capitalized.

3) Fig. 5b: It would be beneficial to have a high magnification view in control areas for comparison.

The white squares in the left part of Figure 5b-e indicate the area that is used for a high magnification view on the right part of each panel. In the high magnification views of lesioned animals, we can see the large areas without healthy NeuroTrace straining and are mainly showing debris and puncta in comparison to the healthy neurons in non-lesion controls.

Figure 5. Ablation of Pf_{FL} but not VAL_{FL} cells before learning impairs the refinement of initial reach direction

(a) Experimental design showing the timeline for surgeries and STT training, and coronal ABA sections showing the injection targets for Pf_{FL} and VAL_{FL} bilateral lesions.

(b) Representative coronal section of VGlut2-Cre mouse injected with AAV expressing taCasp3 - NeuroTrace (yellow) targeted to Pf_{FL} . Dashed-line indicating Pf. High-magnification view from the inset in box.

(c) Representative coronal section of WT-littermate mouse injected with AAV expressing taCasp3 - NeuroTrace (yellow) targeted to Pf_{FL} . Dashed-line indicating Pf. High-magnification view from the inset in box.

4) Fig. 5j-m: Controls do not seem to be consistent with Fig. 1. Any interpretation? This is why Fig. 1 requires statistics.

A possible reason for the difference in controls in Figure 1 and Figure 5 is that in Figure 5, we use both male and female mice, which increased the overall variability of the total group and can account for the differences. In Figure 1, all mice are male. Another possible difference is that the animals in Figure 1 are trained in a different behavioral apparatus that sits under a 2-photon microscope, and the controls in Figure 5 were trained in sound-attenuating behavior boxes in a different room. Because of this, there are various differences in the environment related to noise and temperature that can fluctuate and impact behavior. Ultimately, even though the controls are not completely consistent from the animals in Figure 1 and 5, we do not believe it detracts from the main takeaways in Figure 5 that are related to the effect of pre-learning lesions.